# WORK ZONES CHALLENGE VLM TRAJECTORY PLANNING: TOWARD MITIGATION AND ROBUST AUTONOMOUS DRIVING

## ABSTRACT

Visual Language Models (VLMs), with powerful multimodal reasoning capabilities, are gradually integrated into autonomous driving by several automobile manufacturers to enhance planning capability in challenging environment. However, the trajectory planning capability of VLMs in work zones, which often include irregular layouts, temporary traffic control, and dynamically changing geometric structures, is still unexplored. To bridge this gap, we conduct the *first* systematic study of VLMs for work zone trajectory planning, revealing that mainstream VLMs fail to generate correct trajectories in 68.0% of cases. To better understand these failures, we first identify candidate patterns via subgraph mining and clustering analysis, and then confirm the validity of 8 common failure patterns through human verification. Building on these findings, we propose REACT-Drive, a trajectory planning framework that integrates VLMs with Retrieval-Augmented Generation (RAG). Specifically, REACT-Drive leverages VLMs to convert prior failure cases into constraint rules and executable trajectory planning code, while RAG retrieves similar patterns in new scenarios to guide trajectory generation. Experimental results on the ROADWork dataset show that REACT-Drive yields a reduction of around $3\times$ in average displacement error relative to VLM baselines under evaluation with Qwen2.5-VL. In addition, REACT-Drive yields the lowest inference time (0.58s) compared with other methods such as fine-tuning (17.90s). We further conduct experiments using a real vehicle in 15 work zone scenarios in the physical world, demonstrating the strong practicality of REACT-Drive. Our code and demos are available on https://sites.google.com/view/react-drive.

## 1 INTRODUCTION

Autonomous driving has progressed rapidly, with both academia and industry devoting extensive efforts to vehicles that can operate in urban and highway environments with minimal or no human supervision. Owing to the strong performance of deep learning in pattern recognition and large-scale data processing, significant progress has been achieved in safety (Jahan et al., 2019), trajectory planning (Leon & Gavrilescu, 2021), and visual perception (Chen et al., 2020), with increasing applications in real-world scenarios (Baidu Apollo, 2025; Mercedes-Benz USA, 2025). Although autonomous driving have demonstrated increasingly reliable performance in structured environments such as urban roads and highways, recent incidents highlight their pronounced limitations in complex and dynamic settings such as work zones. Work zone crashes remain a significant safety concern in roadway transportation, with nearly 100,000 incidents occurring annually in the United States and more than 40,000 individuals injured each year (Pennsylvania Department of Transportation, 2019). In 2017, a Tesla Model S Autopilot failed to recognize road signs and crossed a temporary barrier (Autoweek, 2017). In 2023, a Cruise robotaxi entered an active construction site (Los Angeles Times, 2023). In 2025, a Xiaomi SU7 failed to decelerate when approaching a highway construction site under intelligent driving assistance, resulting in three fatalities (CarNewsChina, 2025). A common feature of these accidents is that they all occurred when autonomous vehicles encountered work zones. Work zones are characterized by long-tail and highly dynamic conditions, such as irregular layouts, which pose significant challenges for autonomous driving systems.

Vision Language Models (VLMs) combine strong visual perception and language understanding capabilities. With their advantage in zero-shot transfer, they have been shown to address complex road planning problems in autonomous driving (Zhou et al., 2024). According to recent reports, several automotive companies, including Li Auto Inc. (Li Auto Inc., 2024) and Geely (Geely Holding Group, 2025), have begun integrating VLMs into their autonomous driving systems. However, our study shows that current VLMs are inadequate for trajectory planning in work zone scenarios. For instance, mainstream VLMs like Qwen2.5-VL (Bai et al., 2025) achieve an FDE of only 285.90 on the ROADWork dataset (Ghosh et al., 2024), in contrast to 106.38 on the normal commonsense driving cases in NuScenes (Caesar et al., 2020).

To better understand the abnormal behaviors, we filter out those abnormal scenarios (images) where all VLMs fail to provide the correct path. We then follow a three-step analysis framework to discover the main causes of VLM failures. We first construct scene graphs from different abnormal scenarios. Based on these graphs, we conduct abnormal subgraph mining and candidate merging, then apply clustering and inflection-point analysis to identify 10 representative patterns. Finally, we manually summarize and verify these patterns, which reveal 8 main patterns of VLM failures in work zone scenarios. Based on the summarized abnormal patterns, we propose a two-stage framework: Retrieval-Enhanced And Constraint-verified Trajectory for Driving (REACT-Drive). In the offline stage, failure cases are converted into con-

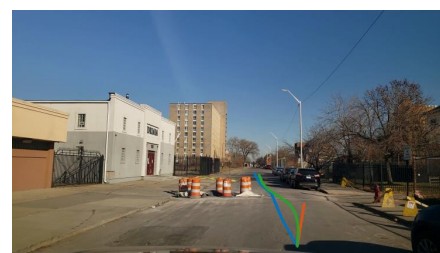

Figure 1: REACT-Drive can effectively generate a correct planning trajectory in complex work zone scenarios. The red line denotes the QWEN2.5's output trajectory, the green line is GT and the blue line denotes Real-REACT's output trajectory.

straint rules expressions and executable trajectory mitigation code. A self-verification mechanism is applied to ensure their usability, resulting in a searchable failure case mitigation code database. In the online stage, we use a retrieval-augmented generation (RAG) pipeline: the current scenario is encoded as a query to retrieve cases matching the failure patterns, and the associated historical mitigation code is then executed. This process guides the autonomous driving system to generate trajectories that comply with safety requirements and traffic rules. Experimental results demonstrate that REACT-Drive improves trajectory planning performance in work zones in both effectiveness and efficiency. For example, when evaluated with Qwen2.5-VL-72B as the backbone model, REACT-Drive achieves an around $3\times$ reduction in average displacement error compared with VLM baselines. Furthermore, we perform physical evaluation with data collected from 15 real-world work zones, which shows that REACT-Drive consistently achieves lower error. We envision that REACT-Drive can shed light on future research in improving work zone driving ability via VLMs.

Overall, we make the following contributions:

- We perform the first evaluation of VLM-based driving systems on the trajectory planning task in work zones, revealing that current VLMs struggle significantly with this task.
- Through scene graph mining and human verification, we conduct abnormal pattern analysis on failure cases and summarize 8 typical failure patterns of VLM-based driving systems. We then propose a two-stage framework REACT-Drive that leverages constraint rules and RAG to reuse mitigation code for safe trajectory planning in new scenarios.
- Evaluations show that REACT-Drive significantly reduces trajectory prediction errors for around $3\times$ and planning overhead to $0.58s$. In addition, we conduct physical experiments in real-world work zone scenarios using data collected from autonomous driving vehicles, which further validate the effectiveness of REACT-Drive.

## 2 RELATED WORK

**Vision Language Models for Autonomous Driving.** Vision Language Models (VLMs) map text and image inputs into a shared embedding space, which provides models with unified perception and natural language reasoning capabilities. This property enables them to achieve broad applicability across multimodal downstream tasks (Khan et al., 2023; Shao et al., 2023; Hu et al., 2021;

Sun et al., 2025; Zhang et al., 2025). In the field of autonomous driving, VLMs also demonstrate strong potential. They integrate language reasoning with driving tasks, which not only enhances interpretability but also improves adaptability in new environments and under unfamiliar traffic rules due to their strong zero-shot generalization ability. Building on these advantages, recent studies explore the application of VLMs to autonomous driving scenarios (Mao et al., 2023; Huang et al., 2024; Sima et al., 2024; Ma et al., 2024; Jiang et al., 2024; Shao et al., 2024; Xing et al., 2025). VLMs for autonomous driving integrate multimodal perception inputs (images, videos, lidar, radar) with textual inputs (user instructions, ego-vehicle states) to support perception, navigation, planning, and decision making (Zhou et al., 2024). In trajectory planning, OpenEMMA (Xing et al., 2025) leverages camera images and textualized histories with pre-trained VLMs to predict trajectories, both adapting open-source foundation models. To address VLMs' limitations in handling precise coordinates, Senna (Jiang et al., 2024) fine-tunes a VLM for scene understanding and high-level decision making, producing meta-actions that the Senna-E2E planner (Jiang et al., 2023) translates into accurate trajectories. Beyond planning, VLMs are also used for scene understanding, question answering, and high-level driving decisions. For instance, DriveLM (Sima et al., 2024) introduces the Graph-VQA task for step-by-step reasoning over multi-view images. Dolphins (Ma et al., 2024) fuses video or image inputs, textual instructions, and control histories for dialogue-based assistance.

**Work Zones in Autonomous Driving.** Compared to the usual urban and highway driving scenarios, work zones present unique challenges for autonomous driving systems. They often involve long-tail and highly dynamic conditions, such as temporary lane shifts, traffic cones, construction vehicles, on-site workers, nighttime lighting interference, and residual lane markings. Previous studies on autonomous driving in work zone scenarios mainly focus on perception. Gumpp et al. (2009) and Mathibela et al. (2013) concentrate on reliable detection and recognition of work zones. Similarly, the CODA dataset (Li et al., 2022) is primarily used for corner case evaluation at the object level, including rare construction-related targets, while the RoSA dataset (Kim et al., 2024) provides segmentation of highway construction zones. In contrast, the **ROADWork** dataset (Ghosh et al., 2024) is the first dataset designed for the task of driving through work zones, which contains 1186 scenarios It provides multi-granularity annotations of work zone elements and further enables the estimation of drivable trajectories from real driving videos for trajectory planning. Therefore, we conduct our evaluation based on the ROADWork dataset.

## 3 VLM Evaluations and Abnormal Pattern Analysis

Based on the ROADWork dataset, we conduct a systematic evaluation on multiple mainstream VLM autonomous driving frameworks. The overall results are shown in Table 1. In this dynamic and highly uncertain environment, the trajectory planning performance of all models degrades significantly. To better analyze the causes of failure, we perform abnormal pattern analysis. As illustrated in Figure 2, the process consists of two steps: (1) Scene Graph Construction, where we build scene graphs for failed cases and establish directional, proximity, and lane-membership relations; and (2) Subgraph Mining & Human Analysis, where we mine abnormal subgraphs, cluster them, and incorporate human verification to refine and attribute the underlying failure mechanisms. Fi-

Table 1: ROADWork failure scenarios. A case is a failure if ADE$> 50px$ and FDE$> 100px$ (The definitions of metrics are shown in Section 5.1); A scenario is a failure if $> 50\%$ of its cases fail.

| Model | #. Failure Scenarios | Percentage (%) |
|---|---|---|
| GPT4o | 822 | 70.37 |
| Qwen2.5-VL | 886 | 75.86 |
| Gemini2.5 | 911 | 80.00 |
| SimLingo | 957 | 81.93 |
| RoboTron-Drive | 895 | 76.63 |
| DriveLM | 902 | 77.23 |

nally, we summarize the abnormal subgraphs into 8 typical patterns, which form the basis of our mitigation framework REACT-Drive.

### 3.1 Scene Graph Construction

Inspired by (Woodlief et al., 2024), we encode the image as a directed scene graph centered on the ego vehicle with its work zone elements. Specifically, we construct the scene graph $G = (V, E)$, where $V$ denotes the nodes of entities related to work zones (derived from auxiliary categories $\mathcal{C}_{\mathrm{aux}}$ and detected work zone categories $\mathcal{C}_{\mathrm{wz}}$) and $E$ represents the relations among these entities.

Figure 2: Pipeline Overview. Abnormal Pattern Analysis builds scene graphs, clusters abnormal sub-graphs, and summarizes patterns; REACT-Drive constructs a mitigation-code database from failure cases and retrieves the right code for new scenarios.

**Node Construction.** To detect elements related to work zones, we first fine-tune a dedicated detector based on Yolov12 (details are provided in Appendix D). Given an input frame, the detector returns an instance set $\{(v_i, b_i)\}$, where each $v_i$ belongs to the work zone category set $\mathcal{C}_{\mathrm{wz}}$ (refer to Table A4), and $b_i = [x_i, y_i, w_i, h_i]$ is its bounding box. We combine these detected nodes with a small set of auxiliary structure nodes: $\mathcal{C}_{\mathrm{aux}} = \{\text{ego}, \text{Left Lane}, \text{Middle Lane}, \text{Right Lane}, \text{Root Road}\}$.

**Edge Construction.** After obtaining the nodes, we establish edge relations for them. The edge set is defined as a collection of directed triplets with relation labels:

$$E = \{(v_i, v_j, r) \mid v_i \in V, \ v_j \in V, \ r \in \mathcal{R}\}. \tag{1}$$

Here $\mathcal{R}$ consists of three mutually disjoint families of relations: $\mathcal{R} = \mathcal{R}_{\mathrm{dir}} \cup \mathcal{R}_{\mathrm{prox}} \cup \mathcal{R}_{\mathrm{lane}}$, where the $\mathcal{R}_{\mathrm{dir}}$ denotes directional relations, $\mathcal{R}_{\mathrm{prox}}$ denotes proximity relations, and $\mathcal{R}_{\mathrm{lane}}$ denotes lane membership relations. To ensure stable orientation judgments across frames, the centers of detection boxes are mapped from pixel coordinates to a meter-based plane centered at the ego vehicle. Let the image center be $(c_x, c_y)$, the pixel-per-meter conversion factor be PPM, and the $i$-th instance with pixel-level bounding box $b_i = (x_i^{\mathrm{px}}, y_i^{\mathrm{px}}, w_i^{\mathrm{px}}, h_i^{\mathrm{px}})$, the relative coordinates are defined as:

$$(x_i, y_i) = \left( \frac{x_i^{\mathrm{px}} + \frac{w_i^{\mathrm{px}}}{2} - c_x}{\mathrm{PPM}}, \ \frac{c_y - \left(y_i^{\mathrm{px}} + \frac{h_i^{\mathrm{px}}}{2}\right)}{\mathrm{PPM}} \right), \tag{2}$$

where the $y$-axis points forward and the ego vehicle is located at $(0, 0)$.

For the *directional relations* $\mathcal{R}_{\mathrm{dir}}$, we consider three types: inFrontOf, toLeftOf, and toRightOf. The direction is determined by the relative position of the source node $v_i$ and the target node $v_j$. Let their positions on the meter-based plane be $\mathbf{p}_i = (x_i, y_i)$ and $\mathbf{p}_j = (x_j, y_j)$, and let the orientation of $v_i$ be $\psi_i$ (if the ego vehicle is the source, then $\psi_{\mathrm{ego}} = 0$).

$$\Delta \mathbf{p}_{i \to j} = \mathbf{p}_j - \mathbf{p}_i, \qquad \theta_{i \to j} = \mathrm{wrap}_{(-\pi, \pi]}\big(\mathrm{atan2}(\Delta y, \Delta x) - \psi_i\big), \tag{3}$$

where $\mathrm{wrap}_{(-\pi, \pi]}$ normalizes any angle to the interval $(-\pi, \pi]$, and $\mathrm{atan2}(\Delta y, \Delta x)$ returns the polar angle of the vector $(\Delta y, \Delta x)$ measured from the global x-axis. Given the relative angle $\theta_{i \to j}$ from $i$ to $j$ at time $t$ and a threshold $\alpha$, define $\mathrm{dir}(v_i, v_j)$ as inFrontOf if $|\theta_{i \to j}| \leq \alpha$, toLeftOf if $\theta_{i \to j} > \alpha$, and toRightOf if $\theta_{i \to j} < -\alpha$.

For the *proximity relations* $\mathcal{R}_{\mathrm{prox}}$ that encode distance, we estimate the meter-level distance from the source node $v_i$ to the target node $v_j$ using a pretrained monocular depth model (MiDaS (Ranftl et al., 2022) in our work), denoted $d_{i \to j}$ (or $d_i$ when the source is the ego vehicle), and instantiate the five relations: near_collision, super_near, very_near, near, visible, with fixed thresholds $[0, 4)$, $[4, 7)$, $[7, 10)$, $[10, 16)$, and $[16, 25)$ meters, respectively.

For the *lane membership relations* $\mathcal{R}_{\mathrm{lane}}$, in the ego-centric frame we assign each foreground node $v_i$ to a virtual lane based on its lateral coordinate $x_i$ with $L$ denoting half of the lane width, setting

lane($v_i$) to Left Lane if $x_i < -L$, to Middle Lane if $|x_i| \leq L$, and to Right Lane if $x_i > L$, and then adding edges ($v_i$, lane($v_i$), isIn) and (lane($v_i$), Root Road, isIn).

Therefore, starting from the ego vehicle, we construct three types of relations. For each foreground node $v_i$ (with class belonging to $\mathcal{C}_{\mathrm{wz}}$ detected by the detector), we sequentially generate directional, proximity, and lane membership edges, namely (ego, $v_i$, $r_{\mathrm{dir}}$), (ego, $v_i$, $r_{\mathrm{prox}}$), and ($v_i$, lane($v_i$), isIn), where $r_{\mathrm{dir}} \in \mathcal{R}_{\mathrm{dir}}$ and $r_{\mathrm{prox}} \in \mathcal{R}_{\mathrm{prox}}$. By combining all the edges, we obtain the scene graph of a frame $G = (V, E)$.

## 3.2 SUBGRAPH MINING & HUMAN ANALYSIS

To structure the failure cases of VLMs in work zones and facilitate human summarization of the final abnormal patterns, we perform subgraph mining on the scene graphs of the abnormal frame set $\mathcal{A}$. Candidate subgraphs are first generated and merged by isomorphism, resulting in a set of prototype subgraphs (prototypes) that serve to cluster the failure cases. The final abnormal clusters and their patterns are then summarized by humans based on these evidences.

**Candidate Subgraph Extraction.** Let the full graph of frame be $G = (V, E)$, where $V$ denotes the node set of the entire graph. To avoid notation conflict, we denote the node set of a candidate subgraph $S$ as $V^S$, which is a subset of $V$ selected according to specific rules (i.e., $V^S \subseteq V$). Specifically, we start from the ego node and perform a depth-limited breadth-first search with depth $D = 2$ on $G$ by traversing only outgoing edges, while expanding only nodes belonging to the work zone category set $\mathcal{C}_{\mathrm{wz}}$:

$$V^S = \{\texttt{ego}\} \cup \left\{ v \in V : \mathrm{dist}_G^+(\texttt{ego}, v) \leq D, \ \mathrm{label}(v) \in \mathcal{C}_{\mathrm{wz}} \right\}, \tag{4}$$

where $\mathrm{dist}_G^+$ denotes the directed distance considering only outgoing edges. Based on this vertex set, we define the candidate subgraph as $S \triangleq G[V^S]$ that is, the vertex-induced subgraph on $V^S$ can be defined as $S = \{(u, v, r) \in E : u, v \in V^S\}$. If $|V^S| < m$ ($m = 3$ in our evaluation), the candidate is discarded. For each image $i$ in dataset $D$, we construct a candidate subgraph $S_i$. The resulting pool of candidate subgraphs (over all images) is $\mathcal{S}_{\mathrm{abn}} \triangleq \left\{ S_i \mid i \in \mathcal{D}, |V^{S_i}| \geq m \right\}$.

**Candidate Merging.** We merge the abnormal candidate set $\mathcal{S}_{\mathrm{abn}}$ through a four-step procedure: (1) Signature-based Bucketing, where candidates with identical structural signatures are grouped; (2) Threshold Gating, which filters pairs that differ substantially in scale; (3) Subgraph Containment Check, which tests relation-preserving subgraph isomorphism; and (4) Union-find Merging, which merges related candidates and selects the smallest subgraph as representative. Implementation details are provided in the Appendix E.

**Cluster.** After candidate merging, we perform clustering on the resulting abnormal subgraphs. The procedure is as follows: first, for each merged subgraph, we extract both structural features (such as statistics of nodes and relations, subgraph size, and average depth) and CLIP visual features (obtained by aggregating instance-level box features within the subgraph). Next, the two types of features are normalized separately and concatenated into a unified representation, on which K-means clustering is applied over all samples. We then compute the sum of squared errors (SSE) for different numbers of clusters $K \in \{2, \dots, K_{\max}\}$ and apply the elbow method (Aranganayagi & Thangavel, 2007) to automatically select the optimal number of clusters. On our data, the knee point appears at $K = 10$, and thus we obtain 10 clusters, which are taken as the initial abnormal clusters.

**Human Verification and Failure Analysis.** While the abnormal pattern analysis pipeline effectively compresses and organizes large sets of failure cases, there is no semantic interpretation of these clusters. To obtain a human-understandable pattern from each cluster, we introduce a human analysis stage. To be specific, we perform two tasks: (i) *failure pattern combination*: we manually verify and combine similar failure patterns. (ii) *semantic summarization*: each pattern is assigned descriptive labels according to the object types, relative spatial relations, and contextual cues.

This process yields a concise and interpretable set of failure patterns, where each pattern corresponds to a systematic weakness of VLM-based planning in work zone scenarios. The outcome bridges the gap between automated discovery and human interpretability, and the resulting failure taxonomy can be directly employed for both quantitative benchmarking and qualitative analysis. The details of these 8 patterns can be referred to Appendix G.

# 4 MITIGATION FRAMEWORK: REACT-Drive

To address those failure cases, we propose the mitigation framework REACT-Drive. The core idea of this framework is to combine retrieval-augmented generation (RAG) with constraint rules, leveraging retrievable failure cases and verifiable constraints to stabilize trajectory planning results. As shown in Figure 2, the overall framework consists of two stages: Failure case mitigation code database construction and RAG-based inference on failure cases.

## 4.1 FAILURE CASE MITIGATION CODE DATABASE CONSTRUCTION

At this stage, our goal is to convert the failure cases in the database into the corresponding constraint rules and executable trajectory planning code (the prompt is provided in Table A3). We then ensure their correctness through self-verification before adding them to the retrieval database.

**Prompt Construction and Constraint Generation .** To obtain the work zone constraints, we input a single-frame image overlapping with the failure trajectories into the VLMs. These failure trajectories serve as a reference for the VLM to infer the expected work zone constraints in the scene. Based on the work zone traffic regulations defined in (Pennsylvania Department of Transportation, 2019), we generate a set of 8 predefined work zone constraint template encoding specific regulations related to work zones. The VLM processes the failure trajectory and image context to determine which constraints apply to the given scenario, such as detouring, maintaining the center of the lane, returning to the original lane after bypassing the work zone, *etc.*These constraints ensure that the vehicle moves safely within the work zone environment (see Appendix H for details).

**Mitigation Code Generation and Self-verification Feedback.** We query VLM to generate mitigation code based on the generated constraints. The generated code consists of two parts: *segment_drivable_mask* and *plan_destination* to handle road mask adjustment and destination planning under work zone constraints (see Appendix F for details).

After obtaining the destination generated by the code, we will employ a smoothing algorithm to plan the trajectory. This trajectory will be generated based on the starting point, destination, drivable road mask, and the work zone information. The process will output a trajectory consisting of 20 discrete points that guide the vehicle from the starting position to the destination.

To verify whether the generated code is consistent with the failure reference cases, we introduce a self-validation feedback stage, which consists of two types of constraints: **(1) Drivability Constraint:** We construct an Euclidean distance transform from the drivable set $\Omega_{\text{drive}}$ defined by the road mask, where $D(\mathbf{x})$ denotes the minimum distance from point $\mathbf{x}$ to $\Omega_{\text{drive}}$. The predicted target point $\mathbf{x}_{\text{pred}}$ is required to satisfy $D(\mathbf{x}_{\text{pred}}) \leq \tau_{\text{road}}$. If this threshold is exceeded, the prediction is regarded as invalid. **(2) Destination Constraint (Predicted Point vs. GT Destination):** The predicted target point is required to remain within a Euclidean distance threshold from the GT target point in the pixel coordinate space $d_{\text{pix}} = \sqrt{(x_{\text{pred}} - x_{\text{gt}})^2 + (y_{\text{pred}} - y_{\text{gt}})^2} \leq \tau$. When either constraint is violated (e.g., the predicted point lies far from the drivable region, or the predicted destination deviates excessively from the GT), the system returns actionable feedback containing numerical errors and visual comparisons, and a retry is triggered. Once verification passes or the maximum number of iterations is reached, the constraint rules and the corresponding code are stored together with the validation record.

## 4.2 RAG-BASED INFERENCE ON FAILURE CASES

After constructing the failure-case mitigation code database, we adopt the RAG on the database. When the vehicle encounters a new scenario, the system first extracts multimodal features of the scene, including images, semantic annotations, and temporal context. These features are then fed into the RAG module to efficiently retrieve similar cases from the database, and the corresponding mitigation code is used to obtain the correct planning.

**Triplet-based Similarity Retrieval.** Finally, for each abnormal prototype $R_g$, we retrieve the Top-$K$ candidates from the normal sample pool using CLIP (Radford et al., 2021) features (with priority given to the temporal prefix of the same sequence, $K \in \{5, 10\}$). For each retrieved normal candi-

date subgraph $N$, we compute a triplet of similarity measures:

$$\text{sim}_{\text{struct}} = \lambda \text{Jacc}\big(\text{set}(L(R_g)), \text{set}(L(N))\big) + (1 - \lambda)\text{Jacc}\big(\text{set}(R(R_g)), \text{set}(R(N))\big),$$

$$\text{sim}_{\text{depth}} = \max\left\{0,\ 1 - \frac{\left|\bar{d}(R_g) - \bar{d}(N)\right|}{3.5}\right\}, \qquad \text{sim}_{\text{bbox}} = \max\left\{0,\ 1 - \frac{\left|\bar{\delta}(R_g) - \bar{\delta}(N)\right|}{\max\{960, 540\}}\right\}.$$

$$(5)$$

Based on these measures, we define a rule for priority re-ranking (without performing isomorphism checks): $\text{non\_indep}(R_g) \iff \min\{\text{sim}_{\text{struct}}, \text{sim}_{\text{depth}}, \text{sim}_{\text{bbox}}\} \geq 0.8$, otherwise $R_g$ is marked as independent, and its priority for human verification is increased.

Once the RAG module confirms the presence of a highly similar failure case, the system directly invokes the stored mitigation code and executes it in the real-time environment. Since these codes are verified through a self-validation mechanism, their correctness and executability can be ensured to a certain extent. In this way, when facing complex scenarios, the vehicle does not need to reason entirely from scratch but can quickly leverage failure experience to improve stability and safety. Meanwhile, abnormal scenarios that do not pass the similarity threshold are directly handled by the reasoning of the VLM-based driving system.

## 5 EVALUATION

### 5.1 EXPERIMENTAL SETUP

**Dataset.** To evaluate the performance of the VLM autonomous driving system, we use the ROAD-Work dataset (Ghosh et al., 2024) as the testing dataset. This dataset collects multiple city-scale driving scenes that contain diverse work zone scenarios. It also provides trajectory annotations that capture normal driving behaviors in work zone environments as the ground truths.

**Models and Baselines.** In this work, we consider 6 representative VLMs: GPT4o, Gemini-2.5, Qwen2.5-72B-VL (hereafter referred to as Qwen2.5), Qwen3-VL-30B-A3B-Thinking (hereafter referred to as Qwen3-reasoning), DriveLM, SimLingo, and RoboTron-Drive. For fair evaluation, the first three VLMs are deployed under the OpenEMMA framework for inference, while the others are evaluated in their original implementations. Second, to further assess the effectiveness of our constraint-rule mechanism, we compare REACT-Drive against two additional baselines: (i) **fine-tune VLM**, which fine-tuned Qwen2.5-72B-VL using QLoRA (Dettmers et al., 2023) directly on ROADWork; and (ii) **VLM (Self) w/o Constraint Rules**, which performs self-prediction without incorporating our constraint rules. These are evaluated alongside our method in the mitigation study (refer to Table 4).

**Metrics.** Following previous work (Sima et al., 2024; Huang et al., 2024; Xing et al., 2025), we adopt three standard open-loop planning metrics. Specifically, Average Displacement Error (ADE) measures the average displacement error between the predicted and ground-truth trajectories, Final Displacement Error (FDE) measures the final displacement error, and Collision Rate (CR) measures the average collision rate across generated trajectories. For all of these metrics, a lower value indicates better performance.

$$\text{ADE} = \frac{1}{T}\sum_{t=1}^{T}\|\hat{\mathbf{p}}_t - \mathbf{p}_t\|_2, \quad \text{FDE} = \|\hat{\mathbf{p}}_T - \mathbf{p}_T\|_2, \quad \text{CR} = \frac{1}{N}\sum_{i=1}^{N}C(\hat{Y}_i) \qquad (6)$$

where $\hat{\mathbf{p}}_t \in \mathbb{R}^2$ is the predicted position at time step $t$, $\mathbf{p}_t \in \mathbb{R}^2$ is the ground-truth position, $T$ is the prediction horizon, $N$ is the number of predicted trajectories, and $C(\hat{Y}_i)$ is an indicator function that equals to 1 if trajectory $\hat{Y}_i$ collides with other agents or obstacles, and 0 otherwise. Noted that, since accurate depth information is not available in ROADWork, we measure ADE and FDE in the image coordinate (pixel) space rather than in meters.

### 5.2 EXPERIMENTAL RESULTS

**VLM Performance on Different Patterns.** Table 2 presents the performance of different VLMs across 8 patterns on the ROADWork dataset, which shows that all 6 models perform unsatisfactorily

Table 2: Effectiveness of different VLMs across 8 patterns (lower is better). Colors indicate relative performance within each metric (ADE/FDE/CR) across the entire table: deep red = worse.

| | P1: Dense drums or cones on sidewalk | | | P2: Encounter dead end road | | | P3: Interference from large work vehicles | | | P4: Lane borrowing through work zone | | |
|---|---|---|---|---|---|---|---|---|---|---|---|---|
| | ADE | FDE | CR | ADE | FDE | CR | ADE | FDE | CR | ADE | FDE | CR |
| **GPT4o** | 115.54 | 218.39 | 0.15 | 198.87 | 463.98 | 0.05 | 99.78 | 184.04 | 0.15 | 118.03 | 242.63 | 0.19 |
| **Qwen2.5** | 121.83 | 203.51 | 0.27 | 231.02 | 524.38 | 0.38 | 111.12 | 209.80 | 0.26 | 120.55 | 248.55 | 0.17 |
| **Qwen3-Reasoning** | 139.06 | 267.89 | 0.14 | 218.61 | 481.93 | 0.09 | 110.64 | 203.62 | 0.04 | 133.26 | 243.53 | 0.10 |
| **Gemini2.5** | 238.10 | 502.04 | 0.10 | 263.73 | 610.16 | 0.03 | 149.20 | 207.62 | 0.25 | 242.68 | 533.70 | 0.17 |
| **DriveLM** | 146.84 | 296.55 | 0.12 | 215.83 | 522.33 | 0.00 | 126.33 | 228.80 | 0.04 | 123.77 | 260.45 | 0.06 |
| **SimLingo** | 280.94 | 437.59 | 0.02 | 419.25 | 693.70 | 0.04 | 283.87 | 349.73 | 0.04 | 329.77 | 435.75 | 0.01 |
| **RoboTron-Drive** | 132.71 | 247.13 | 0.18 | 217.08 | 486.15 | 0.00 | 136.19 | 248.61 | 0.00 | 123.67 | 236.43 | 0.12 |
| **Average** | 167.86 | 310.44 | 0.14 | 252.06 | 540.38 | 0.08 | 145.30 | 233.17 | 0.11 | 170.25 | 314.43 | 0.12 |

| | P5: Lane shift across work zones | | | P6: Overreaction to signs | | | P7: Accelerate through the exit in the work zone | | | P8: Turning through work zone | | |
|---|---|---|---|---|---|---|---|---|---|---|---|---|
| | ADE | FDE | CR | ADE | FDE | CR | ADE | FDE | CR | ADE | FDE | CR |
| **GPT4o** | 148.23 | 306.39 | 0.07 | 118.31 | 215.92 | 0.08 | 105.38 | 163.99 | 0.14 | 153.84 | 353.75 | 0.07 |
| **Qwen2.5** | 167.16 | 328.33 | 0.16 | 96.50 | 140.18 | 0.07 | 83.10 | 145.73 | 0.10 | 235.44 | 486.72 | 0.24 |
| **Qwen3-Reasoning** | 98.78 | 188.36 | 0.11 | 128.84 | 244.12 | 0.05 | 141.29 | 258.31 | 0.11 | 145.84 | 305.12 | 0.14 |
| **Gemini2.5** | 225.04 | 491.55 | 0.01 | 255.28 | 629.73 | 0.13 | 328.79 | 764.47 | 0.04 | 212.15 | 576.28 | 0.16 |
| **DriveLM** | 218.51 | 479.01 | 0.00 | 129.29 | 263.39 | 0.02 | 126.11 | 251.81 | 0.10 | 213.07 | 462.08 | 0.00 |
| **SimLingo** | 348.67 | 555.88 | 0.01 | 254.09 | 374.66 | 0.05 | 286.19 | 458.95 | 0.01 | 413.82 | 630.31 | 0.03 |
| **RoboTron-Drive** | 93.25 | 192.95 | 0.10 | 111.13 | 222.81 | 0.00 | 99.46 | 190.88 | 0.06 | 253.49 | 575.27 | 0.00 |
| **Average** | 185.66 | 363.21 | 0.07 | 156.21 | 298.69 | 0.06 | 167.19 | 319.16 | 0.08 | 232.52 | 484.22 | 0.09 |

in work zones. The prediction errors remain high, with overall ADE around 192.15, FDE close to 371.94, and average collision risk CR about 0.09. These results indicate that existing VLMs struggle to achieve reliable trajectory prediction and safe avoidance in the complex environments of work zones. Among these VLMs, GPT4o achieves the best overall performance with the lowest average FDE of 268.64. In contrast, Gemini2.5 performs the worst overall, with an average FDE as high as 539.44, indicating severe long-horizon errors. Across work zone patterns, difficulty varies notably. P2 and P8 are the hardest cases: P2 has an average FDE of 550.12, reflecting long-term convergence errors; P8 reaches 514.07, showing turning in work zones carries the greatest prediction risk. Qwen3-Reasoning offers slightly more stable performance than Qwen2.5 on several patterns, but still suffers from large prediction errors in work zones. Overall, current VLMs exhibit clear limitations in work zone environments.

**Mitigation Effectiveness and Performance Factors.** Table 4 displays the average performance across the three mitigation settings. We observe that REACT-Drive achieves the best overall results. The fine-tuned version yields an ADE of 207.97, FDE of 384.31, and CR of 0.11. The self-reasoning VLM without constraint rules further reduces the ADE and FDE to 201.09 and 350.75, respectively, with CR dropping to 0.03. Our REACT-Drive significantly lowers the ADE and FDE to 54.73 and 101.64, while keeping CR at a relatively low level of 0.04. In terms of pattern-specific mitigation, our method shows the most notable improvements in P1, P3, P6, and P7, where the collision risk is reduced to 0.00 in all four cases. In contrast, P4 and P8 show slightly increased CR, but their FDE is significantly reduced. This is because these two patterns sometimes occur in turning scenarios, the cones and barriers are inherently very close to the lane center, so any trajectory that closely follows the ground-truth path will also lie very close to the obstacle masks in the image space, making slight overlaps under the pixel-level CR metric. This effect is largely due to the limitation of the current work-zone dataset, which lacks precise 3D geometric information. At present, we can only approximate CR by checking whether the 2D trajectory intersects with 2D obstacle masks. Please refer to the examples of P5 and P8 in Figure A2. When the destination is correctly outputted, the trajectory may be inevitably closed to work zone objects. Furthermore, we investigate whether different VLM models will influence the mitigation effective-

Table 3: Comparison of VLMs on mitigation tasks using REACT-Drive.

| **Model** | ADE ↓ | FDE ↓ | CR ↓ |
|---|---|---|---|
| GPT4o | 54.73 | 101.64 | 0.04 |
| Qwen2.5 | 86.46 | 124.67 | 0.07 |

Figure 3: Inference time comparison across different methods.

ness. Table 3 reports the comparison between GPT4o and Qwen2.5 on mitigation tasks. Overall, GPT4o demonstrates stronger performance across all three metrics. Specifically, it achieves an ADE of $54.73$, FDE of $101.64$, and CR of $0.04$, which are consistently lower than Qwen2.5.

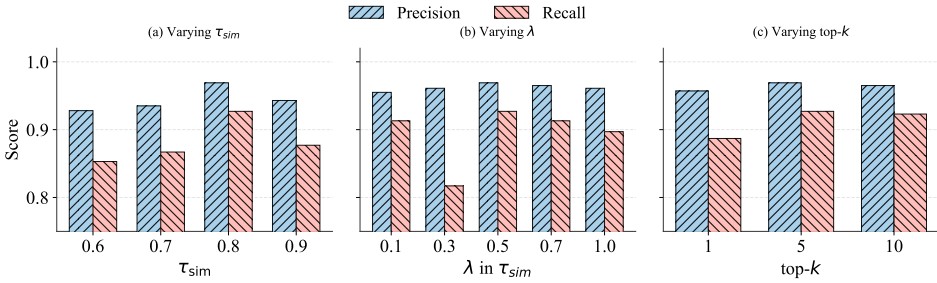

Figure 4: Ablation studies on key hyperparameters. (a) Varying similarity threshold $\tau_{\mathrm{sim}}$. (b) Varying structural similarity threshold. (c) Varying top-$k$.

**Ablation of Different Hyperparameters.** To validate the rationality of our hyperparameter choices in REACT-Drive, we conduct a systematic ablation study on three key hyperparameters in the RAG-based failure-case retrieval module described in Section 4: (1) the similarity threshold $\tau_{\mathrm{sim}}$, which decides whether the current scene is similar enough to a historical failure case; (2) the $\lambda$ in the structural similarity term $\mathrm{sim}_{\mathrm{struct}}$, balancing geometric structure and other features; and (3) the number of retrieved candidates Top-$K$, i.e., how many similar samples are retrieved for each prototype. We randomly sample 300 failure and 100 normal cases from the ROADWork dataset, and for each configuration we report failure-pattern coverage (Recall) and retrieval precision (Precision).

As shown in Figure 4, increasing the similarity threshold $\tau_{\mathrm{sim}}$ from $0.6$ to $0.8$ improves both Precision (from $0.928$ to $0.969$) and Recall (from $0.853$ to $0.927$), suggesting that a moderately higher threshold better suppresses noisy matches while enhancing failure-pattern coverage. However, pushing the threshold to $0.9$ reduces Recall (to $0.877$) and slightly lowers Precision, as overly strict filtering begins to reject genuinely similar cases. Hence, $\tau_{\mathrm{sim}} = 0.8$ yields a well-balanced operating point.

The results on the $\lambda$ in the structural similarity term $\mathrm{sim}_{\mathrm{struct}}$ show that very small weights (e.g., $0.1$ or $0.3$) make the model less sensitive to geometric-structural differences and reduce Recall, while very large weights (e.g., $0.7$ or $1.0$) overemphasize structural cues and also degrade Recall. In contrast, setting the $\lambda$ to $0.5$ yields the best trade-off, achieving the highest Precision ($0.969$) and Recall ($0.927$), which suggests that a balanced use of structural and other features is crucial for capturing stable failure patterns.

The ablation on Top-$K$ shows that retrieving only Top-1 candidate leads to a lower Recall ($0.887$) due to insufficient coverage, whereas $K = 5$ gives the highest Precision and Recall ($0.969$ and $0.927$). Increasing to $K = 10$ slightly reduces both metrics, indicating that a larger candidate pool brings limited gains while introducing more noisy matches. Therefore, we adopt $\tau_{\mathrm{sim}} = 0.8$, a $\mathrm{sim}_{\mathrm{struct}}$ weight of $0.5$, and Top-$K = 5$ as the default configuration, which maximizes failure-pattern coverage while keeping incorrect matches low.

Table 4: Mitigation results by failure pattern (lower is better). We consider Qwen2.5 as the VLM. Colors are relative to the per-pattern average from Table 2: blue = better, red = worse.

| Pattern | fine-tune VLM | | | VLM (Self) w/o Constraint Rules | | | REACT-Drive (VLM + Constraint Rules) | | |
|---|---|---|---|---|---|---|---|---|---|
| | ADE↓ | FDE↓ | CR↓ | ADE↓ | FDE↓ | CR↓ | ADE↓ | FDE↓ | CR↓ |
| P1 | 145.17 | 328.16 | 0.12 | 177.91 | 367.36 | 0.00 | 62.55 | 121.53 | 0.00 |
| P2 | 257.78 | 532.96 | 0.14 | 246.16 | 414.34 | 0.00 | 52.51 | 110.44 | 0.02 |
| P3 | 230.17 | 389.73 | 0.20 | 176.25 | 318.64 | 0.03 | 61.09 | 118.40 | 0.00 |
| P4 | 90.60 | 226.47 | 0.04 | 210.72 | 466.83 | 0.00 | 21.56 | 37.51 | 0.17 |
| P5 | 227.46 | 392.14 | 0.26 | 130.28 | 278.36 | 0.12 | 84.21 | 146.17 | 0.03 |
| P6 | 177.48 | 399.02 | 0.05 | 146.70 | 275.27 | 0.08 | 46.47 | 83.38 | 0.00 |
| P7 | 187.04 | 274.06 | 0.02 | 157.67 | 215.42 | 0.00 | 35.21 | 68.70 | 0.00 |
| P8 | 236.10 | 330.87 | 0.04 | 363.03 | 469.79 | 0.02 | 74.21 | 127.02 | 0.12 |
| Average | 207.97 | 384.31 | 0.11 | 201.09 | 350.75 | 0.03 | 54.73 | 101.64 | 0.04 |

**Efficiency.** Figure 3 compares inference time across different baselines. Our method achieves the lowest latency ($< 1$s per scene), substantially faster than all other approaches. In contrast, fine-tuning large-scale VLMs incurs significant overhead ($\sim 18$s), making them impractical for real-time deployment. Other baselines such as GPT4o and RoboTron-Drive show moderate latency ($\sim 5$s and $\sim 3$s), while SimLingo maintains relatively low cost but is still slower than our constraint-enhanced design. These results highlight that our approach not only improves planning robustness in work zones but also provides clear efficiency advantages.

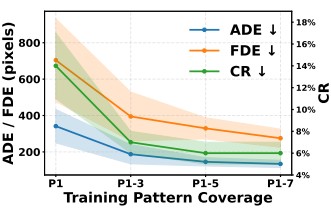

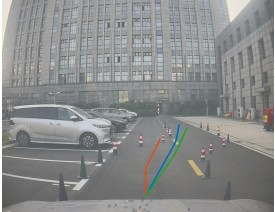

| Model | ADE ↓ | FDE ↓ | CR ↓ |
|---|---|---|---|
| GPT4o | 190.81 | 396.00 | 0.08 |
| Finetune-VLM | 184.56 | 374.84 | 0.07 |
| DriveLM | 244.16 | 488.68 | 0.02 |
| SimLingo | 238.83 | 441.21 | 0.02 |
| RoboTron-Drive | 220.98 | 453.32 | 0.03 |
| REACT-Drive (Ours) | 62.2 | 107.35 | 0.05 |

Figure 5: Pattern coverage experiments.

Figure 6: An example of physical experiments.

Table 5: Physical experiment results.

**Transferability.** To further evaluate the transferability of REACT-Drive, we conduct a set of physical experiments. Specifically, we collect 23 real-world work-zone scenarios from local driving environments that are geographically different from ROADWork, covering a total of 200 diverse work-zone images (example: Figure 6) with different types of cones, drums, and other work-zone elements. For each image, two authors jointly annotate ground-truth trajectories to serve as evaluation references. This setup allows us to directly test whether our method can generalize the learned abnormal patterns to unseen physical scenes. Table 5 reports the overall ADE, FDE, and CR averaged over all 200 images, while a detailed breakdown by six representative failure patterns (P1, P2, P3, P5, P7, P8) is provided in Appendix Table A2. Compared with baselines such as GPT4o, Finetune-VLM, DriveLM, SimLingo, and RoboTron-Drive, REACT-Drive consistently achieves lower ADE, FDE, and CR. On average, REACT-Drive attains ADE of 62.20, FDE of 107.35, and CR of 0.05, significantly outperforming all baselines (for example, GPT4o obtains ADE of 190.81, FDE of 396.00, and CR of 0.08). These results demonstrate that integrating abnormal-pattern mining with constraint rules substantially improves robustness when deploying to different work-zone environments in real-world settings. We note that collecting and annotating real-world work-zone data is costly in both time and financial resources, which currently constrains the scale of this physical evaluation.

In addition, we test whether our patterns can effectively address unseen abnormal patterns by progressively increasing the coverage of patterns. Specifically, we vary the coverage of patterns during the training phase and evaluate their generalization ability on pattern 8. Figure 5 summarizes the results. When only pattern 1 is used, the model struggles with ADE 341.57, FDE 703.26, and CR 0.14. As more patterns are involved, performance improves monotonically with FDE to 275.07 when covering 7 out of 8 patterns. These results indicate that pattern diversity plays a key role in enabling generalization: the more representative abnormal patterns are included, the better the model can handle unseen work zone cases.

## 6 CONCLUSION

This paper conducts a systematic study of the limitations of VLMs in autonomous driving trajectory planning on work zones. By systematically analyzing failure cases, we summarize 8 abnormal patterns. Building on this analysis, we propose a retrieval-augmented mitigation method (REACT-Drive), which converts failure cases into constraint rules and executable planning code that are integrated into the trajectory generation process. Evaluation on the ROADWork dataset shows that REACT-Drive significantly improves prediction performance. Moreover, we conduct physical evaluations to show the effectiveness of REACT-Drive. Our work reveals the notable deficiencies of VLMs in complex and dynamic work zone environments and presents a feasible solution pathway to enhance the robustness of autonomous driving systems in work zones and other safety-critical scenarios. We also provide an extended discussion in the Appendix B on key practical concerns, including perception robustness, trajectory metrics in pixel space, coverage of adverse conditions, and the scalability and scenario scope of the failure-case code database.

REPRODUCIBILITY STATEMENT

To facilitate reproducibility, we clearly describe the models, training, and inference procedures in the methodology and experimental sections. We will open-source all key components and provide anonymously downloadable code, with the access link available on the anonymous project page. The released demo, code, and instructions together ensure that readers can reproduce the experimental results.

ETHICS STATEMENT

This study does not involve human subjects or personal privacy data. All data are obtained from public datasets and real-vehicle experiments collected under strict safety measures. The research objective is to enhance the safety of autonomous driving systems in construction zones, and the proposed methods are designed in compliance with traffic regulations while minimizing potential risks. No conflict of interest exists in this work.

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

## A  THE USE OF LARGE LANGUAGE MODELS

LLMs did not play a significant role in the ideation or writing of this paper. Any incidental use was limited to non-substantive tasks (e.g., grammar or formatting checks) and did not influence the research design, analysis, or textual content.

## B  LIMITATIONS

Although we propose the retrieval-augmented mitigation method (REACT-Drive), achieving significant improvements in work zones, several limitations remain: 1. **Limited scenario coverage:** This work does not systematically cover other long-tail scenarios, such as work zones under extreme weather or nighttime driving. Although the dataset may contain a few such samples, we have not conducted a dedicated analysis; 2. **Limited dataset:** The evaluation is conducted only on the ROADWork dataset and physical data collected by us, without including other work zone datasets. This limitation mainly arises from the scarcity of accessible data. In the future, we plan to construct larger-scale and more diverse work scene datasets; 3. **Lack of real-vehicle deployment:** our work have not been deployed REACT-Drive on real autonomous vehicles, which leaves a potential gap from real-world road environments. This is because conducting such experiments with autonomous vehicles in work zones would be extremely dangerous.

## C  DISCUSSION OF KEY CONCERNS REGARDING REACT-Drive

**Perception errors from Yolov12 / MiDaS and safety risks.** In our implementation, YOLOv12 and MiDaS are employed as convenient off-the-shelf perception modules to construct the scene graph, but REACT-Drive is not tied to these specific models. In realistic autonomous driving systems, higher-accuracy perception outputs from LiDAR, millimeter-wave radar, stereo vision, or sensor-fusion stacks are already reliably available, so potential errors from YOLOv12 or MiDaS should be understood as an artifact of this particular instantiation rather than a fundamental limitation of the framework. Moreover, REACT-Drive is designed to be highly modular: any stronger detector, depth estimator, or multi-sensor fusion module can seamlessly replace the current perception components without modifying the reasoning and planning pipeline. This model-agnostic design allows the proposed error analysis and reactive planning framework to directly benefit from future advances in perception.

**Trajectory metrics in pixel space.** In our experiments, ADE and FDE are reported in pixel space, which indeed imposes certain limitations on the evaluation. This choice is mainly constrained by the ROADWork dataset, which does not provide accurate GPS or 3D depth information needed to convert image-space trajectories into meter-level trajectories. For the physical experiments, we further use the camera intrinsic and extrinsic parameters to estimate an approximate pixel-to-meter conversion and evaluate the trajectories at real-world scale. Under this calibration-based approximation, the final FDE of Pattern 5 is estimated to be about 1.49 meters. Given the observed scene geometry and the expected operational conditions, this level of deviation is considered acceptable for the evaluated scenarios.

**Coverage of extreme weather and nighttime conditions.** The current experiments do not systematically include extreme-weather or nighttime scenarios, mainly due to limitations of the ROAD-Work dataset. Although ROADWork is currently the most comprehensive open-source work-zone

dataset with trajectory annotations, it does not contain diverse adverse-weather or low-light conditions. Recent studies have shown that VLMs maintain reasonable robustness under challenging visual conditions such as illumination changes and moderate adverse weather Xu et al. (2024); Yang et al. (2024). When stronger upstream perception modules are available, for example multi-sensor fusion, enhanced low-light imaging, or nighttime perception models, REACT-Drive can naturally benefit from these improvements because of its modular and model-agnostic design. To provide an initial demonstration, we additionally evaluate three nighttime scenarios similar to pattern 1, 5 (Figure A1a shows an example) and 8 using the recently released Waymo dataset Ettinger et al. (2021). The resulting (AvgDET, AvgFDE, CR) metrics for the three scenarios are (24.08, 39.44, 0), (26.78, 36.93, 0) and (40.73, 82.38, 0), respectively, indicating that the proposed framework retains reasonable performance under nighttime conditions. It is also worth noting that extremely low-visibility conditions, such as dense fog, blizzards, or severe nighttime glare, often fall outside the operational design domain of most autonomous driving systems and pose significant risks even for human drivers. Such scenarios should be handled through system-level safety strategies, for example, operation restrictions, active degradation, or controlled fallback maneuvers, rather than relying solely on the trajectory planning module.

**Failure-pattern quantitative analysis.** To provide a clearer and more systematic characterization of the typicality and coverage of the eight failure patterns, we compute their empirical distribution over all successfully migrated failure cases in the test set. As summarized in Table A1, P1, P5, and P7 are the most common categories, while the other patterns still account for a non-trivial fraction of failure cases. In addition, approximately 7% of all test cases (roughly 200 cases) are directly fed into the VLMs without matching any of the above patterns because their similarity scores during inference fall below the threshold.

Table A1: Distribution of the eight failure patterns over all 2,664 successfully migrated failure cases in the ROADWork test set.

| Pattern ID | Pattern category | Ratio (%) |
|---|---|---|
| P1 | Dense drums or cones on sidewalk | 22.1 |
| P2 | Encounter dead-end road | 12.7 |
| P3 | Interference from large work vehicles | 8.2 |
| P4 | Lane borrowing through work zone | 8.4 |
| P5 | Lane shift across work zones | 11.6 |
| P6 | Overreaction to signs | 8.3 |
| P7 | Accelerate through the exit in the work zone | 16.2 |
| P8 | Turning through work zone | 12.5 |

**Scalability and maintenance of the failure-case code database.** Since the failure-case database is constructed and queried offline, its scalability mainly depends on the efficiency of the retrieval module. REACT-Drive adopts a lightweight RAG-based vector retrieval mechanism, and the scene-graph features it uses are low-dimensional. As a result, even when the database grows, the retrieval latency increases only slightly and does not affect real-time performance. In addition, the structure and traffic rules of work zones are inherently limited, so the number of meaningful failure patterns and their corresponding mitigation codes is theoretically bounded, and the database will not grow without limit. When deploying in different regions, the database can be partitioned by area, with each region maintaining its own failure-case database, which further prevents unnecessary expansion. Because the entire database is maintained offline, outdated or poorly performing codes can also be periodically pruned based on retrieval statistics or validation results. This design enables REACT-Drive to scale to larger deployments while keeping the retrieval efficient and the case library well curated.

**Scope of scenarios: focus on work zones.** We focus on work zones because they are among the most dangerous, accident-prone, and challenging long tail scenarios for autonomous driving. In the United States alone, there are approximately 100,000 work zone crashes each year, and several high profile autonomous driving incidents have occurred in construction areas. This indicates that current systems perform well on structured roads but still exhibit noticeable failures in work zones, which is the core motivation of our study.

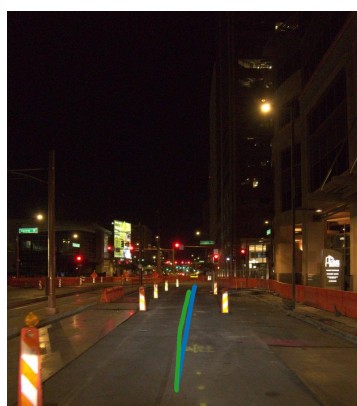 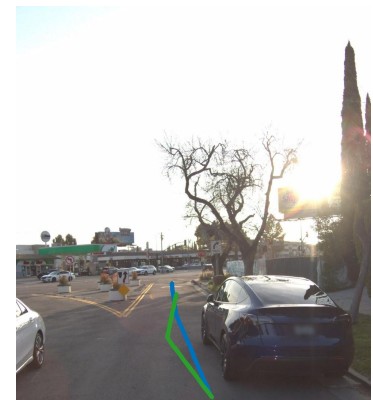

(a) REACT-Drive in a nighttime work-zone scenario. This case is similar to our pattern 5. The green line is GT and the blue line denotes Real-REACT's output trajectory.

(b) REACT-Drive in another real-world scenario. This case is similar to our pattern 4. The green line is GT and the blue line denotes Real-REACT's output trajectory.

Figure A1: Qualitative examples of REACT-Drive under nighttime and other real-world scenarios.

Table A2: Physical experiments on real-world work-zone scenarios across 6 failure patterns.

| Model | P1: Dense drums or cones on sidewalk | | | P2: Encounter dead end road | | | P3: Interference from large work vehicles | | |
|---|---|---|---|---|---|---|---|---|---|
| | ADE | FDE | CR | ADE | FDE | CR | ADE | FDE | CR |
| GPT4o | 143.21 | 294.98 | 0.05 | 276.06 | 575.79 | 0.06 | 125.58 | 267.79 | 0.00 |
| Finetune-VLM | 104.75 | 206.26 | 0.02 | 336.47 | 657.53 | 0.06 | 109.62 | 217.66 | 0.00 |
| DriveLM | 165.40 | 341.53 | 0.03 | 368.69 | 728.15 | 0.06 | 167.84 | 351.33 | 0.00 |
| SimLingo | 184.89 | 327.17 | 0.02 | 358.07 | 705.67 | 0.00 | 179.89 | 314.67 | 0.00 |
| RoboTron-Drive | 152.30 | 328.40 | 0.05 | 355.20 | 710.50 | 0.00 | 155.60 | 338.20 | 0.00 |
| REACT-Drive | 52.88 | 104.84 | 0.00 | 61.46 | 145.32 | 0.00 | 64.74 | 112.38 | 0.00 |
| Model | P5: Lane shift across work zones | | | P7: Accelerate through the exit in the work zone | | | P8: Turning through work zone | | |
| | ADE | FDE | CR | ADE | FDE | CR | ADE | FDE | CR |
| GPT4o | 152.52 | 280.43 | 0.25 | 161.36 | 335.68 | 0.00 | 286.12 | 621.34 | 0.10 |
| Finetune-VLM | 129.27 | 252.82 | 0.21 | 109.44 | 217.10 | 0.00 | 317.83 | 697.67 | 0.10 |
| DriveLM | 226.78 | 436.79 | 0.04 | 161.36 | 335.68 | 0.00 | 374.86 | 738.60 | 0.00 |
| SimLingo | 203.05 | 360.87 | 0.08 | 143.33 | 248.59 | 0.00 | 363.76 | 690.28 | 0.00 |
| RoboTron-Drive | 173.40 | 320.30 | 0.08 | 128.90 | 302.10 | 0.03 | 360.50 | 720.40 | 0.00 |
| REACT-Drive | 64.04 | 87.40 | 0.15 | 61.33 | 92.77 | 0.00 | 68.76 | 101.39 | 0.15 |

To the best of our knowledge, we are the first to conduct a systematic evaluation of VLM-based planning in work zones. Our results reveal severe failures of existing VLMs in this setting, which highlights the necessity of the proposed method. Moreover, the proposed framework is not limited to work zones and can be extended to other long tail scenarios in autonomous driving by redefining the failure patterns and mitigation codes for the target scenario. For example, we further evaluate our framework on three representative long-tail scenarios from the Waymo dataset Ettinger et al. (2021) that similar with our patterns 4 (Figure A1b shows an example ), 7 and 8. These cases include lane borrowing to bypass parked cars, interactions with agents approaching from an adjacent lane, and turning behaviors at intersections that create additional collision risks. Under these scenarios, our method achieves (ADE, FDE, CR) = (30.24, 46.07, 0), (35.70, 59.90, 0) and (56.36, 87.46, 0), outperforming a GPT-4o baseline, which produces (44.14, 88.05, 0), (57.56, 124.10, 0) and (71.20, 142.71, 0.5) under identical settings. These results demonstrate that our framework can also handle related long tail scenarios beyond work zones.

## D   DETAILS OF TRAINING YOLOV12

Our work fine-tunes the Yolov12 model on the ROADWork dataset, focusing on the work zone object categories listed in Table A4. The training configuration is set to 100 epochs with a batch size of 16 and an initial learning rate of 0.01.

## E   IMPLEMENTATION OF CANDIDATE MERGING

For the abnormal candidate set $\mathcal{S}_{\mathrm{abn}}$, we perform a four-step merging.

The first step is *Signature-based Bucketing*. For each candidate $S_i$, we compute a structural signature and store it into a bucket:

$$\sigma(S_i) = \left( L(S_i),\ R(S_i),\ |V^{S_i}|,\ |E^{S_i}| \right),$$

where $L(S_i) = \{\mathrm{label}(v) : v \in V^{S_i}\}$ and $R(S_i) = \{\mathrm{rel}(e) : e \in E^{S_i}\}$ denote the multisets of node labels and edge relations, respectively (with duplicates preserved). In subsequent steps, pairwise comparisons are performed only within the same signature bucket.

The second step is *Threshold Gating*, which further filters candidate pairs within each signature bucket. For any subgraph $S_i$, we define:

$$\bar{d}(S_i) = \frac{1}{|V^{S_i}|} \sum_{v \in V^{S_i}} \mathrm{depth}(v), \qquad \bar{\delta}(S_i) = \frac{1}{|V^{S_i}|} \sum_{v \in V^{S_i}} \left\| \mathrm{ctr}(b_v) - c \right\|_2,$$

where $\bar{d}(S_i)$ is the average depth of nodes and $\bar{\delta}(S_i)$ is the average pixel radius (the distance to the image center $c$). A pair $(S_i, S_j)$ proceeds to the next step only if $\left| \bar{d}(S_i) - \bar{d}(S_j) \right| \leq 1.0$, $\left| \bar{\delta}(S_i) - \bar{\delta}(S_j) \right| \leq 150$ px, otherwise it is skipped. This gating eliminates subgraphs that share the same signature but differ significantly in scale, without affecting topology, and it substantially reduces the number of comparisons required.

The third step is *Subgraph Containment Check*. For each gated pair $(S_i, S_j)$, we perform a directed and relation-preserving subgraph isomorphism containment test. Let $S_{\min}$ be the subgraph with fewer nodes. If there exists an injection $\phi : V(S_{\min}) \to V(S_{\max})$ such that $\mathrm{label}(v) = \mathrm{label}(\phi(v))$, $\quad (u,\ v,\ r) \in E(S_{\min}) \Rightarrow (\phi(u),\ \phi(v),\ r) \in E(S_{\max})$, then we consider $S_{\min} \preceq S_{\max}$.

The final step is *Union-Find Merging*. Using a union–find structure, we merge candidate pairs that satisfy the containment relation. After processing all buckets, this yields several connected components (clusters). For each cluster $g$, we select $R_g = \arg\min_{S \in g} |V^S|$ as the representative, corresponding to the smallest subgraph that provides the minimal evidence.

## F   IMPLEMENTATION OF ROAD MASK SEGMENTATION AND DESTINATION PLANNING

---

**def segment_drivable_mask(road_mask, workzone_bboxes)**

This function is responsible for segmenting the drivable road mask based on work zone constraints. The input to this function includes the original road mask and the work zone element bounding boxes and depths. Based on these constraints, the function adjusts the road mask by blocking undrivable regions accordingly. For example, if detour_side is set to left, the function will set the right side of the road as undrivable to avoid the right work zone. The output is a drivable road mask.

---

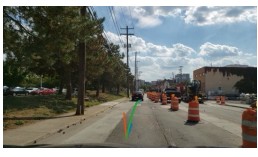 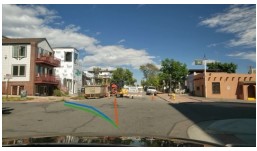 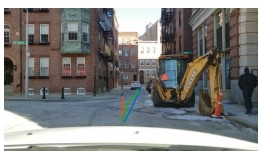 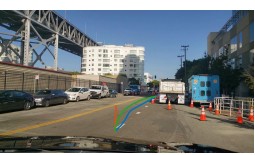

**P1: Dense drums or cones on sidewalk:** This pattern needs to follow **1** constraint rule: *follow the lane center.*

**P2: Encounter dead end road:** This pattern needs to follow **1** constraint rule: *turn to avoid work zone.*

**P3: Interference from large work vehicles:** This pattern needs to follow **2** constraint rules: *follow the lane center, detour to bypass the work zone.*

**P4: Lane borrowing through work zone:** This pattern needs to follow **2** constraint rules: *return the origin lane after bypassing the work zone, detour to bypass the work zone.*

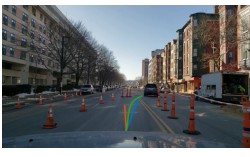 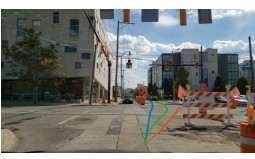 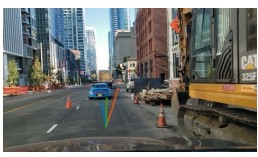 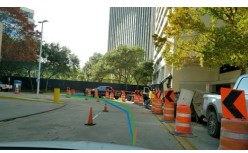

**P5: Lane shift across work zones:** This pattern needs to follow **2** constraint rules: *cross the work zone, return center line after crossing work zone.*

**P6: Overreaction to signs:** This pattern needs to follow **2** constraint rules: *follow the sign, return center line after crossing work zone.*

**P7: Accelerate through the exit in the work zone:** This pattern needs to follow **2** constraint rules: *follow the front car, follow the lane center.*

**P8: Turning through work zone:** This pattern needs to follow **2** constraint rules: *cross the work zone, follow the lane center.*

Figure A2: Eight failure patterns (P1-P8) with per-pattern rule constraints.

---

**`def plan_destination(road_mask, workzone_bboxes)`**

This function utilizes the modified drivable road mask and work zone information to determine a valid destination point based on the constraints. For example, if *return_to_original_lane_after_work_zone* is set to True, the function places the destination near the original lane after the vehicle has bypassed the work zone. The function outputs the final destination coordinates (x, y), ensuring the vehicle follows a safe and effective trajectory that respects work zone constraints.

---

## G  DETAILS OF PATTERNS

The details of patterns are shown on Figure A2. The red line denotes the QWEN2.5's output trajectory, the green line is GT and the blue line denotes Real-REACT's output trajectory.

## H  PROMPT FOR CONSTRAINT RULES AND MITIGATION CODE GENERATION

Table A3: The prompt template for generating work zone constraints and mitigation code. The code (in red text) and the constraint samples (in blue text) are to be filled.

**Background**:
You are a failure for writing autonomous-driving **constraint rules and planning code**. You are extremely good at modeling driving scene constraints and translating them into executable code. You SHOULD first provide your step-by-step thinking for solving the task.

**Task**:
You will be given:
1) ONE front-view image that contains a green failure trajectory and a visible failure destination point.
2) A partially pre-filled combined constraint template (Work zone Constraints).
3) A Ground Truth (GT) annotated image showing green trajectory points, work zone boundaries, and road mask overlay. Use this GT information to understand the expected behavior and generate appropriate constraint rules and planning code.
You MUST return **two blocks in order, with NO extra commentary**:
(A) A completed Constraints in JSON.
(B) Executable Python code containing two functions: segment_driveable_mask and plan_destination.

**Constraints Template**:
The JSON you must complete is as follows. Fill every "UNKNOWN" slot without altering the structure.
```
{
  "constraints": {
    "no_cross_workzone":  "UNKNOWN", // Determines if crossing through the workzone is al-
lowed ("no" to bypass, "yes" to cross)
    "detour_side":  "UNKNOWN", // Specifies which side to detour when bypassing the workzone
("left", "right", "none")
    "return_to_original_lane_after_workzone":  "UNKNOWN", // Specifies if the vehicle
should return to the original lane after bypassing the workzone ("True" | "False")
...  }
}
```

**Python Code Requirements**:
You must output two functions:
1) def segment_driveable_mask(original_road_mask, workzone_info): This function should apply depth-aware road mask cutting with workzone constraints. Based on the constraints such as detour_side, it modifies the road mask to block or allow the appropriate regions, for example: If detour_side is left, the right side of the road should be marked as undriveable to avoid the workzone.
2) def plan_destination(driveable_road_mask, workzone_info): This function should calculate the destination based on the modified road mask and constraints, such as: - If return_to_original_lane_after_workzone is True, the function should place the destination near the original lane after bypassing the workzone.

**Expected Results**:
- A completed JSON constraint rules instance based on the template above.
- Full Python implementations of segment_driveable_mask and plan_destination.
- No placeholders. No extra text outside JSON and Python blocks.

Table A4: The category set $\mathcal{C}_{\mathrm{wz}}$ for work zone elements.

| drum | cone | work vehicle | ttc sign | tubular marker |
|---|---|---|---|---|
| barricade | barrier | vertical panel | fence | worker |

