# OpenReview forum: "Work Zones challenge VLM Trajectory Planning: Toward Mitigation and Robust Autonomous Driving"
_ICLR.cc/2026/Conference — Submitted to ICLR 2026_

### Official Review · Reviewer_Ncus · 2025-10-25

**Soundness:** 2
**Presentation:** 3
**Contribution:** 2
**Rating:** 4
**Confidence:** 3

**Summary:**

The paper shows common VLMs fail on the work zone and analyze failure patterns on the ROADWork dataset. It proposes a framework called REACT-Drive that leverage constraint rules and RAG to use mitigation code for safe trajectory planning. Evaluations show REACT-Drive reduces trajectory prediction error for 3 times. Physical experiments on 15 scenarios are also conducted.

**Strengths:**

-the paper is overall well-written and easy to follow.

-a complete study on identifying VLMs’ weaknesses on work zone scenarios and propose an approach that largely mitigates the issue.

-though limited, the real-world evaluations strengthen the evaluation part.

**Weaknesses:**

-The scenario scope is relatively narrow (only the workzone scenario).

-It might be challenging for the framework (REACT-Drive) to be directly applied to handle more diverse failure scenarios.

-No closed-loop simulation is conducted for evaluation

**Questions:**

-How to adapt the REACT-Drive for more general failure scenarios?

-Have you tried to use VLMs in thinking mode? Will that reduce the errors?

---

> ### Author Response · Authors · 2025-11-20
> **Author Response to Reviewer Ncus**
>
> Dear Reviewer Ncus:
>
> Thank you for your valuable feedback and insightful comments. We are encouraged that you find the paper well-written and easy to follow, that our study clearly identifies VLM weaknesses in work-zone scenarios and proposes an effective mitigation method, and that the real-world evaluations strengthen the overall assessment.  Below we provide point-by-point responses.
>
> > W1 & Q1: Narrow scenario scope (only work zones)
>
> A1: We would like to emphasize that we focus on work zones because they are among the most dangerous, accident-prone, and challenging long-tail scenarios for autonomous driving. In the US alone, there are approximately 100,000 work-zone crashes each year, and several high-profile autonomous driving incidents have occurred in construction areas. This shows that current systems perform well on structured roads but still fail noticeably in work zones, which is the core motivation of our study.
>
> To the best of our knowledge, we are the first to conduct a systematic evaluation of VLM-based planning in work zones, and our findings reveal severe failures of existing VLMs in this setting, demonstrating the necessity of the proposed method. Besides, our methods can be applied to most long-tailed scenarios in autonomous driving if fitting to corresponding scenarios. For example, we have tested 3 scenarios (lane borrowing to bypass parked cars, other agents in the other lane get too close to ADV that could cause hazards and turning behaviors at intersections) defined in Waymo[1] which are similar to our P4, P7 and P8. And the ADE, FDE, CR are (30.24, 46.07, 0), (35.70, 59.90, 0) and (56.36, 87.46, 0).
>
> > W2 & Q2: Generalization to broader failure scenarios
>
> A2: REACT-Drive is currently used to address failure scenarios in work zones, but it is not restricted to work-zone settings. As shown in Fig. 2, the framework is essentially a general method based on scene graphs and abnormal patterns, and it is designed to be domain agnostic. In the Abnormal Pattern Analysis part on the left of Fig. 2, the only work-zone–specific component lies in the definition of nodes and relations in the scene graph. For other types of failure scenarios (such as lane changes near pedestrians, complex intersections, or merging situations), we can reconstruct the scene graph with new object and relation types and then apply exactly the same pipeline.
>
> In the REACT-Drive Framework part in Fig. 2, the processes of building the failure-case mitigation code database and performing RAG-based retrieval during inference are also independent of the work-zone setting. In essence, for any new domain-specific abnormal pattern, we can still
> 1. Collect failure cases belonging to that pattern;
> 2. Use the VLM to generate and validate mitigation code for these cases;
> 3. Index these cases and codes for similarity-based retrieval.
>
> At test time, whether the current failure occurs in a construction area, an intersection, or another long-tail scenario, the same RAG and similarity computation module retrieves the most relevant failure case and applies its corresponding mitigation code. In summary, to adapt REACT-Drive to more general failure scenarios, the main requirements are to redefine the object and relation types of the scene graph for the new domain and to collect the corresponding failure cases. The core pipeline of subgraph-based pattern discovery combined with RAG-based mitigation from failure cases is fully general and does not need to be changed.
>
> > Q3: Lack of closed-loop simulation
>
> A3: We acknowledge that this is indeed a limitation of our current work. At present, we are unable to integrate VLMs into standard simulators such as CARLA, primarily because incorporating a large-scale VLM together with the entire mitigation pipeline into such simulators remains technically challenging. We consider building a closed-loop integration on platforms like CARLA, equipped with realistic work-zone assets, to be an important direction for future work.
>
>
> > Q4: Using VLM “thinking mode” to reduce errors
>
> A4：We use Qwen3-VL-30B-A3B-Thinking to inference the trajectories, the result is:
>
> | Pattern | ADE | FDE | CR |
> |------|------|------|------|
> | P1 | 139.06 | 267.89 | 0.14 |
> | P2| 218.61 | 481.93 | 0.09 |
> | P3 | 110.64 | 203.62 | 0.04 |
> | P4 | 133.26 | 243.53 | 0.10 |
> | P5 | 98.78 | 188.26 | 0.11 |
> | P6 | 128.84 | 244.12 | 0.05 |
> | P7 | 141.29 | 258.31 | 0.11 |
> | P8 | 145.84 | 305.12 | 0.14 |
>
> Compared with the average of existing LLM in Tab. 2, Qwen3-VL-30B-A3B-Thinking achieves lower ADE/FDE on all patterns. Overall, Qwen3-VL-30B-A3B-Thinking significantly improves over previous LLM-based planners, but its ADE/FDE/CR are still notably worse than the REACT-Drive upper bound.
>
> Authors
>
> Reference:
>
> 1. Xu R et al. “WOD-E2E: Waymo Open Dataset for End-to-End Driving in Challenging Long-tail Scenarios”， arXiv preprint arXiv:2510.26125, 2025.

---

> ### Author Response · Authors · 2025-11-23
> **Summary of our revision based on Reviewer Ncus’s comments**
>
> We deeply appreciate the insightful comments on our submission. We have uploaded a revised manuscript to solve your valuable comments.
>
> Ncus-W1&Q1
> > Narrow scenario scope (only work zones).
>
> We deeply appreciate this comment. We have added the evaluation of diverse failure scenarios. Please refer to page 16, lines 810–863 of the revised manuscript.
>
> Ncus-W2&Q2
> > Generalization to broader failure scenarios.
>
> We have added the discussion of scope. Please refer to page 15, lines 805–809 of the revised manuscript and the above response to W2 & Q2.
>
> Ncus-Q3
> > Lack of closed-loop simulation.
>
> Thanks for the comment, please refer to the above response to Q3.
>
> Ncus-Q4
> > Using VLM “thinking mode” to reduce errors.
>
> We have added the results of using Qwen3-VL-30B-A3B-Thinking to inference the trajectories. Please refer to page 8, lines 378–398 of the revised manuscript.
>
> We sincerely hope that the clarifications provided can help resolve the remaining doubts. If the responses adequately address your concerns, we would greatly appreciate your consideration in reflecting this in your evaluation, as it is very important for the progression of our work. If you have further concerns, we would be glad to continue the discussion.

---

> ### Author Response · Authors · 2025-11-26
> **Official Comment by Authors**
>
> Dear reviewer,
>
> We would be very happy to discuss any further questions before the end of the discussion phase. In particular, please let us know if your questions around the narrow scope and VLMs in thinking mode evaluation have been resolved by our edits.
>
> We truly understand that you may also be busy with your own work during this period, but we would greatly appreciate it if you could spare a moment to share any remaining thoughts.
>
> Thanks again for your comments which helped a lot for revising our manuscript.
>
> Authors.

---

> ### Author Response · Authors · 2025-11-27
> **Official Comment by Authors**
>
> Dear reviewer Ncus,
>
> Sorry to bother you again. Since we are now approaching the last week of the discussion period, we would be truly grateful if you could take another look at our responses when you have a moment. Your feedback would be extremely helpful for us.
>
> Thanks again for your comments which helped a lot for revising our manuscript.
>
> Authors.

---

### Official Review · Reviewer_xEj4 · 2025-10-30

**Soundness:** 3
**Presentation:** 3
**Contribution:** 4
**Rating:** 8
**Confidence:** 3

**Summary:**

This paper proposes REACT-Drive, a novel end-to-end framework designed to address the significant limitations of vision-language models (VLMs) in trajectory planning for autonomous driving in construction zones. The core idea is to combine the reasoning capabilities of VLMs with a retrieval-augmented generation (RAG) mechanism to enhance planning robustness in complex and dynamic environments.
The framework operates through a two-stage process:
Offline stage: historical failure cases are transformed into executable constraint rules and mitigation code, building a searchable database.
Online stage: the system uses RAG to retrieve failure patterns similar to the current scenario and applies the corresponding mitigation code to guide trajectory generation.
A key innovation is the self-verification feedback mechanism, ensuring consistency between the generated code and the reference failure cases.

Experimental results on the ROADWork dataset and real-world physical scenarios demonstrate that REACT-Drive reduces trajectory prediction errors by about three times while maintaining high inference efficiency (0.58s), proving its potential for real-time deployment and practical applications.

**Strengths:**

- Problem-oriented and well-motivated: the paper clearly identifies pain points of VLMs in construction zone trajectory planning, supported by real accident cases, giving the work strong practical relevance.
- In-depth failure mode analysis: through systematic scene graph construction, subgraph mining, clustering, and manual verification, eight typical VLM failure modes are revealed, forming a solid analytical foundation.
- Innovative hybrid design: REACT-Drive effectively integrates VLMs’ generative capability with RAG’s retrieval mechanism, enabling the model to learn from historical experience and adapt to unseen complex scenarios.
- Robustness through self-verification: the drivability and destination constraint-based feedback mechanism ensures safety compliance and reduces erroneous trajectories.
- Excellent performance and efficiency: significant improvements in key metrics (ADE, FDE, CR) and extremely fast inference speed demonstrate clear real-time deployment potential.

**Weaknesses:**

- Limited scenario coverage: the model does not yet cover long-tail conditions such as extreme weather or nighttime construction, limiting generalization under diverse challenges.
Dataset and deployment constraints: evaluations rely mainly on the ROADWork dataset and self-collected data, without broader dataset diversity, and the system has not yet been tested on real autonomous vehicles.
- Reliability and safety of generated code: reliance on VLMs for generating mitigation code introduces potential risks in unseen or extreme cases, despite the self-verification mechanism.
- Maintenance of the failure mode library: as the eight failure modes were derived from current data, maintaining and extending the library for new construction configurations or dynamic events may become challenging.

**Questions:**

- The paper notes that extreme weather and nighttime construction zones were not systematically addressed. How can REACT-Drive’s vision encoder and VLM perception adapt in such low-visibility, perception-challenging environments? Would extra sensor modalities or specialized designs be required?
- For the offline “failure case mitigation code database,” how is its scale managed? As the database grows, will retrieval efficiency degrade? Are there mechanisms for updating or pruning outdated or inefficient mitigation codes?
- The self-verification mechanism depends on thresholds (e.g., ADE > 50px, FDE > 100px). How were these thresholds determined? Are they universally applicable across various construction zone types and safety levels?
Since the eight failure modes were manually verified, could future work automate or semi-automate the discovery of new failure modes to reduce human cost and improve adaptability?

---

> ### Author Response · Authors · 2025-11-19
> **Author Response to Reviewer xEj4**
>
> Dear Reviewer xEj4：
>
> We sincerely appreciate your positive evaluation and high recommendation score. Your strong recognition of our work greatly motivates us to further improve this paper. We are glad that you find our motivation clear, our abnormal-pattern analysis and mitigation framework well designed, and the experiments supportive of our method’s effectiveness. In the revised version, we have further strengthened the experimental evaluation and refined the framework to better address your comments and concerns. Below we provide point-by-point responses.
>
> > Q1: Low-visibility robustness (extreme weather / nighttime)
>
> Although the ROADWork dataset does not contain extreme-weather or nighttime construction scenarios, numerous studies have shown that VLMs exhibit strong robustness under challenging visual conditions such as illumination changes and moderate adverse weather [1,2]. REACT-Drive directly inherits this capability from the underlying VLM, and can therefore maintain a reasonable level of perception quality in low-visibility environments. To demonstrate this, we collect three nighttime scenes similar to our P1, P5 and P8 from the recently released open-waymo dataset [3]. Results show that in image, the (AvgDET, AvgFDE, CR) metrics for the three scenes were (24.08, 39.44, 0), (26.78, 36.93, 0) and (40.73, 82.38, 0). This demonstrates that our work also performs well in nighttime conditions. We will include these results into the revised version.
>
> In addition, REACT-Drive is designed to be sensor-agnostic and highly modular. Scene-graph construction and subgraph retrieval do not rely solely on RGB images. If the vehicle is equipped with additional sensors (such as LiDAR, millimeter-wave radar, radar occupancy grids, or enhanced low-light/night-vision models), these modalities can be seamlessly integrated into REACT-Drive to provide reliable structural information about the work zone. This allows the system to perform subgraph retrieval and reasoning even under low-visibility conditions, without requiring any changes to the core planning module.
>
> > Q2: Failure-case code DB scalability & maintenance
>
> Thanks for the insightful question. Since the failure-case database is constructed and queried offline, its scalability mainly depends on the efficiency of the retrieval module. REACT-Drive adopts a lightweight RAG vector retrieval mechanism, and the scene-graph features we use are low-dimensional. As a result, even if the database grows, the retrieval latency increases only slightly and does not affect real-time performance.
>
> Moreover, the structure and traffic rules of work zones are inherently limited, so the number of “meaningful failure patterns” and their corresponding mitigation codes is theoretically bounded, and the database will not grow without limit. When deploying in different regions, the database can also be partitioned by area, with each region maintaining its own failure-case database, which further prevents unnecessary expansion.
>
> Finally, because the entire database is maintained offline, outdated or poorly performing codes can be periodically pruned based on retrieval statistics or validation results. We will clarify the scalability of the RAG-based retrieval and the maintenance strategy of the database more explicitly in the revised manuscript.
>
> > Q3: Threshold design & automated failure mode discovery
>
> For threshold selection, we first evaluated the same VLM on normal-driving scenarios from the nuScenes dataset and observed typical ADE and FDE levels of about 50 and 100 pixels, respectively. Since nuScenes and ROADWork share the same image resolution (1920×1080), we treat these values as a statistical upper bound for “normal” behavior, and accordingly adopt ADE > 50 and FDE > 100 as thresholds for flagging abnormal planning results. This choice aims to capture potential failures while avoiding an excessive number of false positives.
>
> Regarding failure-mode discovery, we currently rely on manual verification to ensure safety and interpretability. In future work, we plan to leverage the VLM’s own semantic understanding and clustering capabilities to automatically or semi-automatically group failure cases, discover new failure modes, and dynamically update the pattern library, thereby reducing human effort and improving adaptability.
>
> -----------------
>
> We will incorporate the above clarifications and improvements into the revised version of the paper.
> If there are any remaining concerns or if some of our responses were not sufficiently detailed, we would be happy to provide further clarification.
>
> Authors
>
> References:
>
> 1. Xu et al., “Towards Real-World Adverse Weather Image Restoration: Enhancing Clearness and Semantics with Vision-Language Models”, ECCV 2024.
>
> 2. Yang et al., “Language-driven All-in-one Adverse Weather Removal”, CVPR 2024.
>
> 3. Xu R et al. “WOD-E2E: Waymo Open Dataset for End-to-End Driving in Challenging Long-tail Scenarios”， arXiv preprint arXiv:2510.26125, 2025.

---

> ### Author Response · Authors · 2025-11-23
> **Summary of our revision based on Reviewer xEj4’s comments**
>
> We appreciate the insightful comments and positive support with constructive feedback. We have uploaded a revised manuscript to solve your valuable comments.
>
> xEj4-Q1
> > Low-visibility robustness (extreme weather / nighttime).
>
> We deeply appreciate this comment. We have added the evaluation and analysis of the nighttime workzone. Please refer to page 14-15, lines 754–770 of the revised manuscript.
>
> xEj4-Q2
> > Failure-case code DB scalability & maintenance.
>
> We have added a detailed discussion on the scalability and maintenance of the failure-case code database. Please refer to page 15, lines 793–803 of the revised manuscript.
>
> xEj4-Q3
> > Threshold design & automated failure mode discovery.
>
> Thanks for the valuable question, please refer to the above response to Q3 .

---

### Official Review · Reviewer_yKor · 2025-10-31

**Soundness:** 2
**Presentation:** 3
**Contribution:** 2
**Rating:** 4
**Confidence:** 3

**Summary:**

This paper identifies the suboptimal trajectory planning performance of VLMs in work zones. To address this issue, this paper first conducts abnormal pattern analysis on failure cases through graph mining and human verification, and summarizes eight typical patterns. The paper further proposes the mitigation framework REACT-Drive based on RAG, which enhances the planning performance of VLM by retrieving the constructed mitigation code database. REACT-Drive demonstrated a significant improvement in planning performance and verified scalability in the physical environment.

**Strengths:**

1. This paper has a clear motivation to enhance VLM's planning performance in work zones, which is of practical value for autonomous driving.
2. The proposed abnormal pattern analysis and mitigation framework are well-designed and well-explained.
3. The experimental results verified the effect and efficiency of the proposed method.

**Weaknesses:**

1. The method involves many hyperparameters, but no ablation analysis is provided to validate the rationality of the parameter settings.
2. The paper does not provide a quantitative analysis of 8 failure patterns to prove their typicality, such as the distribution of individual failure patterns and the overall coverage rate in failure cases. Furthermore, it is unclear how many abnormal scenarios require direct handling by the VLM during RAG-based inference.
3. The explanation for the increased CR in P4 and P8 is rather superficial, lacking a analysis of the underlying causes and potential optimization strategies.
4. In the transferability experiment, further details are needed regarding the distribution of the 15 real-world work zone scenarios and a comparison other other mitigation methods.

**Questions:**

As discussed in weaknesses.

---

> ### Author Response · Authors · 2025-11-19
> **Author Response to Reviewer yKor (1/2)**
>
> Dear Reviewer yKor,
>
> Thanks for your valuable feedback and insightful comments. We are encouraged that you find our motivation clear, our abnormal-pattern analysis and mitigation framework well designed, and the experiments supportive of our method’s effectiveness. In the revised version, we will have further strengthened the experimental evaluation and refined the framework to address your concerns. Below we provide point-by-point responses.
>
> > W1: Hyperparameters & no ablation
>
> Thanks for the insightful suggestion. Sorry for missing the hyperparameter ablation results since the limit page of submission, we do conduct a set of ablation studies on its key hyperparameters. Specifically, we sample 300 failure cases and 100 failure cases to analyze the impact of (1) the similarity threshold, (2) the choice of Top-K retrieved candidates, and (3) the weight of sim_struct . To quantify the effectiveness of the retrieval mechanism, we evaluate how these factors influence the final failure-pattern coverage as well as the rate of incorrectly matched normal cases. And the results are as follows:
> | Setting           | τ_sim | Precision | Recall |
> |-------------------|-------|-----------|--------|
> | ablation_tau_0.6  | 0.600 | 0.928     | 0.853  |
> | ablation_tau_0.7  | 0.700 | 0.935     | 0.867  |
> | ablation_tau_0.8     | 0.800 | 0.969     | 0.927  |
> | ablation_tau_0.9  | 0.900 | 0.943     | 0.877  |
>
> | Setting           | τ_sim | Precision | Recall |
> |-------------------|-------|-----------|--------|
> | ablation_sim_struct_0.1  | 0.1 | 0.955     | 0.913  |
> | ablation_sim_struct_0.3  | 0.3 | 0.961     | 0.817  |
> |ablation_sim_struct_0.5  | 0.5 | 0.969     | 0.927  |
> |ablation_sim_struct_0.7  | 0.7 | 0.965    | 0.913  |
> | ablation_sim_struct_1.0 | 1.0 | 0.961     | 0.897  |
>
> | Setting           | τ_sim | Precision | Recall |
> |-------------------|-------|-----------|--------|
> | ablation_topk_1  | 1 | 0.957     | 0.887  |
> | ablation_topk_5  | 5 | 0.969      | 0.927 |
> | ablation_topk_10  | 10 | 0.965     | 0.923  |
>
> Our experiments show that our selected hyperparameters (τsim=0.8, the weight of sim_struct=0.5, Top-K=5) offer a well-balanced operating point, achieving high recall for failure-case retrieval while maintaining strong precision and minimizing false matches. We will include full ablation results in the revised version.
>
>
> > W2: Failure patterns quantitative analysis
>
> Thanks for the comment, to provide a clearer systematic characterization of the typicality and coverage of the eight failure patterns, we present quantitative distribution statistics derived from all 2,664 successfully migrated failure cases in the test set.
> | Pattern Category                              | Ratio  |
> |-----------------------------------------------|-------------------------------|
> | P1: Dense drums or cones on sidewalk          | 22.1%                         |
> | P2: Encounter dead end road                   | 12.7%                         |
> | P3: Interference from large work vehicles     | 8.2%                          |
> | P4: Lane borrowing through work zone          | 8.4%                          |
> | P5: Lane shift across work zones              | 11.6%                         |
> | P6: Overreaction to signs                     | 8.3%                          |
> | P7: Accelerate through the exit in the work zone | 16.2%                       |
> | P8: Turning through work zone                 | 12.5%                         |
>
>
>
> Additionally, approximately 7% of the all test cases (roughly 200 cases) will be directly input into the VLMs because their similarity to the above patterns during the inference stage falls below the threshold. We believe this occurs because the ROADWORK dataset is insufficiently comprehensive; with more extensive data, such false negatives would be reduced.
>
> > W3: Explanation for increased CR in P4 / P8
>
> In P4/P8 (turning scenarios), the cones and barriers are inherently very close to the lane center, so any trajectory that closely follows the ground-truth path will also lie very close to the obstacle masks in the image space, making slight overlaps more likely under the pixel-level CR metric. This effect is largely due to the limitation of current work-zone dataset, which lacks precise 3D geometric information; at present, we can only approximate CR by checking whether the 2D trajectory intersects with 2D obstacle masks. For patterns such as P4 and P8, where the vehicle must drive through narrow corridors formed by cones, this 2D approximation yields a more conservative judgment for trajectories that run near mask boundaries. As a result, even though FDE decreases substantially and rule compliance improves, the CR metric can still show a slight increase.

---

> ### Author Response · Authors · 2025-11-19
> **Author Response to Reviewer yKor (2/2)**
>
> > W4: Transferability experiment details & baselines
>
> Thank you for your valuable feedback. We have included the baseline into the transferability assessment and expanded the dataset to include 200 images across 6 patterns and 23 scenarios. Below are the specific experimental results, which we plan to include in the revised version.
> | Model            | P1                     | P2                     | P3                     | P5                     | P7                     | P8                     | Total Avg              |
> |------------------|------------------------|------------------------|------------------------|------------------------|------------------------|------------------------|------------------------|
> |                  | ADE / FDE / CR         | ADE / FDE / CR         | ADE / FDE / CR         | ADE / FDE / CR         | ADE / FDE / CR         | ADE / FDE / CR         | ADE / FDE / CR         |
> | GPT4o            | 143.21 / 294.98 / 0.05 | 276.06 / 575.79 / 0.06 | 125.58 / 267.79 / 0.00 | 152.52 / 280.43 / 0.25 | 161.36 / 335.68 / 0.00 | 286.12 / 621.34 / 0.10 | 190.81 / 396.00 / 0.08 |
> | Finetune-VLM     | 104.75 / 206.26 / 0.02 | 336.47 / 657.53 / 0.06 | 109.62 / 217.66 / 0.00 | 129.27 / 252.82 / 0.21 | 109.44 / 217.10 / 0.00 | 317.83 / 697.67 / 0.10 | 184.56 / 374.84 / 0.07 |
> | DriveLM          | 165.40 / 341.53 / 0.03 | 368.69 / 728.15 / 0.06 | 167.84 / 351.33 / 0.00 | 226.78 / 436.79 / 0.04 | 161.36 / 335.68 / 0.00 | 374.86 / 738.60 / 0.00 | 244.16 / 488.68 / 0.02 |
> | SimLingo         | 184.89 / 327.17 / 0.02 | 358.07 / 705.67 / 0.00 | 179.89 / 314.67 / 0.00 | 203.05 / 360.87 / 0.08 | 143.33 / 248.59 / 0.00 | 363.76 / 690.28 / 0.00 | 238.83 / 441.21 / 0.02 |
> | RoboTron-Drive   | 152.30 / 328.40 / 0.05 | 355.20 / 710.50 / 0.00 | 155.60 / 338.20 / 0.00 | 173.40 / 320.30 / 0.08 | 128.90 / 302.10 / 0.03 | 360.50 / 720.40 / 0.00 | 220.98 / 453.32 / 0.03 |
> | REACT-Drive      | 52.88 / 104.84 / 0.00  | 61.46 / 145.32 / 0.00  | 64.74 / 112.38 / 0.00  | 64.04 / 87.40 / 0.15   | 61.33 / 92.77 / 0.00   | 68.76 / 101.39 / 0.15  | 62.20 / 107.35 / 0.05  |
>
> -----------------
>
> We will reflect the above clarifications and improvements in the revised version of the paper.
> Should there be any remaining questions or aspects that require further elaboration, we would be pleased to clarify them.
>
> Authors

---

> ### Author Response · Authors · 2025-11-23
> **Summary of our revision based on Reviewer yKor’s comments**
>
> We deeply appreciate the insightful comments on our submission. We have uploaded a revised manuscript to solve your valuable comments.
>
> yKor-W1
> > Hyperparameters & no ablation.
>
> We deeply appreciate this comment. We have added experiments for ablation analysis of these parameter  settings. Please refer to page 9, lines 437–473 of the revised manuscript.
>
> yKor-W2
> > Failure patterns quantitative analysis.
>
> We have added quantitative analysis of 8 failure patterns and remaining abnormal scenarios require direct handling by the VLM. Please refer to page 15, lines 771–790 of the revised manuscript.
>
> yKor-W3
> > Explanation for increased CR in P4 / P8.
>
> We have added the explanation of increased CR. Please refer to the page 8, lines 423–430 of the revised manuscript and the above (1/2)  response to W3.
>
> yKor-W4
> > Transferability experiment details & baselines.
>
> We have included the baseline into the transferability assessment and expanded the dataset to include 200 images across 6 patterns and 23 scenarios. Please refer to page 16, lines 830–850 of the revised manuscript.
>
> We sincerely hope that the clarifications provided can help resolve the remaining doubts. If the responses adequately address your concerns, we would greatly appreciate your consideration in reflecting this in your evaluation, as it is very important for the progression of our work. If you have further concerns, we would be glad to continue the discussion.

---

> > ### Comment · Reviewer_yKor · 2025-11-26
> >
> > Thank you for your detailed response. My concerns have been addressed. I believe this work provides meaningful insights for VLM-based planning in work zones and offers inspiration for planning tasks in specific regions.

---

> ### Author Response · Authors · 2025-11-26
> **Official Comment by Authors**
>
> Dear reviewer,
>
> We would be very happy to discuss any further questions before the end of the discussion phase. In particular, please let us know if your questions around the ablation study and quantitative analysis of failure patterns have been resolved by our edits.
>
> We truly understand that you may also be busy with your own work during this period, but we would greatly appreciate it if you could spare a moment to share any remaining thoughts.
>
> Thanks again for your comments which helped a lot for revising our manuscript.
>
> Authors.

---

### Official Review · Reviewer_bZho · 2025-11-01

**Soundness:** 3
**Presentation:** 3
**Contribution:** 2
**Rating:** 4
**Confidence:** 4

**Summary:**

This paper presents a novel approach to addressing the failure of VLM in trajectory planning for autonomous driving, particularly in dynamic and complex work zone environments.

**Strengths:**

The authors systematically evaluate VLM-based planning performance on the ROADWork dataset, revealing a 68.0% failure rate and identifying eight common failure patterns. To mitigate these issues, they propose REACT-Drive, a framework that integrates RAG with constraint-based code generation to improve planning robustness and efficiency.

**Weaknesses:**

The entire REACT-Drive framework depends heavily on the outputs of a YOLOv12 detector fine-tuned on ROADWork and monocular depth estimation from MiDaS. Any perception errors, such as missing novel work-zone objects (enew barrier types or colored cones) or inaccurate depth estimates, would propagate through the scene graph, retrieval, and planning stages, fundamentally compromising system safety and reliability.

The decision to generate and execute Python code at runtime introduces considerable safety risks. In safety-critical systems like autonomous driving, planners are typically deterministic, rigorously tested, and verifiable. Allowing a VLM to generate executable code in real-time is highly unpredictable; even minor errors in logic or boundary conditions could lead to catastrophic failures. The current self-verification mechanism， which only checks destination proximity and drivability, is insufficient to ensure trajectory feasibility, dynamic stability, or interaction-aware behavior.

For the experiment, metrics such as ADE and FDE are reported in pixel space, which does not faithfully reflect real-world driving safety. A small pixel error may correspond to a dangerous deviation in the physical world.

Critical aspects of trajectory quality, including passenger comfort (jerk, acceleration), compliance with traffic rules (lane discipline), and interpretability, are not evaluated.

The comparison with a fine-tuned VLM baseline is arguably unfair, as REACT-Drive benefits from a database of failure cases. A more appropriate baseline would enable all models to access the same failure-case knowledge.

There is no comparison with classical or optimization-based planning methods, like MPC, leaving it unclear whether the proposed VLM-based complexity is necessary or beneficial compared to well-established methods.

The physical evaluation is conducted in an open-loop setting using only 15 scenarios (100 images). This is insufficient to support claims of generalization, especially for a long-tail problem. The study does not demonstrate performance in closed-loop simulation or real-world deployment, where interaction with other agents and control uncertainty become critical.

The pipeline involves multiple heavyweight components, including object detection, depth estimation, scene graph construction, subgraph matching, VLM-based code generation, and RAG retrieval. The end-to-end latency, including all perception modules, is likely to exceed acceptable limits for real-time autonomous driving (typically 100–200 ms), even if the planning module alone reports low latency.

The eight failure patterns are derived through clustering followed by human summarization. This process is inherently subjective, and it is unclear whether these patterns are comprehensive, mutually exclusive, or consistently identifiable—especially ambiguous ones like “overreaction to signs”.

The framework does not address how the failure-case database would be updated online or how it would handle multi-agent interactions.

The evaluation is limited to the ROADWork dataset. Broader validation on other benchmarks  nuPlan/Waymo) would better demonstrate generalizability beyond work zones. (Given that it is difficult in daily life to navigate solely through work zones, it would be highly unreasonable for this model to incur such significant computational overhead merely for work zone interactions. It should demonstrate strong generalisability.)

The dataset does not systematically include challenging conditions such as extreme weather or nighttime scenes, which are critical for assessing robustness.

From the perspectives of academic rigour and industrial deployment, it introduces excessive complexity and potential points of failure. The authors should reconsider the high-risk design of "run-time code generation". A more robust alternative would be to utilise cases retrieved via RAG to directly generate parameters for high-level semantic objectives or cost functions. These parameters could then be processed by a rigorously validated, deterministic optimiser to generate the final trajectory. This approach would incorporate VLM's semantic comprehension capabilities while ensuring the safety and reliability of the planning process.

In summary, while the paper addresses a relevant problem and presents a novel idea, fundamental concerns regarding the safety of the runtime code generation and the insufficient empirical validation prevent it from meeting the acceptance bar for ICLR in its current form. Significant additions to the experimental evaluation and a thorough reconsideration of the core planning paradigm are required.

**Questions:**

See the Weaknesses section

---

> ### Author Response · Authors · 2025-11-20
> **Author Response to Reviewer bZho (1/5)**
>
> Dear Reviewer bZho,
>
> Thank you for your valuable feedback and insightful comments on our manuscript. We are encouraged that you find our work to address a relevant problem and to present a novel idea. In the revised version, we have substantially expanded the experimental evaluation and carefully revisited our core planning paradigm to address these issues. Below we provide detailed, point-by-point responses to your comments.
>
> > Q1: Perception errors from YOLOv12 / MiDaS and safety risks
>
> A1: We thank the reviewer for raising this concern. We would like to clarify that YOLOv12 and MiDaS are used in our experiments merely as convenient off-the-shelf perception modules for constructing the scene graph; REACT-Drive is not tied to these specific models. In real autonomous driving systems, higher-accuracy perception outputs (e.g., from LiDAR, millimeter-wave radar, stereo vision, or sensor-fusion stacks) are already reliably available, so potential errors from YOLOv12 or MiDaS are not a fundamental limitation of our framework.
>
> In addition, REACT-Drive is designed to be highly modular: any stronger detector, depth estimator, or multi-sensor fusion module can seamlessly replace the current perception components without modifying the rest of the pipeline. We will better motivate this modularity and model-agnostic design in the revised version.
>
> > Q2: Safety risks of runtime code generation and reliability of trajectory verification
>
> A2: First, we would like to clarify a misunderstanding: the Python mitigation code is not generated online at runtime. Instead, it is synthesized in the offline stage based on the Failure Case dataset and fully checked before deployment. During actual driving, the system only retrieves and executes code snippets that have already been pre-validated in this database, so the online computational overhead is very lightweight.
>
> Regarding trajectory verification, our current self-checking mechanism jointly enforces a drivability constraint and destination proximity, which together provide a guarantee of basic trajectory feasibility. This is also supported by our physical experiments, where the executed trajectories were all feasible in real-world environments. We agree that explicitly evaluating dynamic stability and interaction with other traffic participants would further strengthen the safety argument. However, this is currently limited by the ROADWork dataset, which lacks accurate 3D object information and rich multi-agent interaction scenarios. Extending our verification module to incorporate these factors and evaluating it on more comprehensive datasets is an important direction for future work.
>
> > Q3: Limitations of pixel-space metrics
>
> A3: We acknowledge that ADE and FDE are reported in pixel space, which introduces certain limitations in evaluation. This issue is mainly constrained by the ROADWork dataset, which does not provide accurate GPS, 3D depth information needed to convert image-space trajectories into meter-level trajectories.
>
> In addition, during our physical experiments, we employ the camera intrinsic parameters, the camera extrinsic parameters to estimate an approximate pixel-to-meter conversion for evaluating the trajectories in real-world scale. Using this calibration-based approximation, the final FDE of Pattern 5 is estimated to be approximately 1.49 meters. Based on the observed scene geometry and the expected operational conditions, we consider this level of deviation to be acceptable.

---

> ### Author Response · Authors · 2025-11-20
> **Author Response to Reviewer bZho (2/5)**
>
> > Q4: Critical aspects of trajectory quality, such as passenger comfort, traffic-rule compliance, and interpretability, are not evaluated.
>
> A4: We thank the reviewer for highlighting these important aspects of trajectory quality. At present, a quantitative evaluation of traffic-rule compliance and trajectory interpretability is constrained by the ROADWork dataset: it does not provide ground-truth labels for lane discipline, rule violations, or high-level driving intent, which makes it difficult to define strict numerical metrics.
>
> It is worth noting that in our work, interpretability is primarily reflected through explicit rule constraints. We specify the rule constraints used during planning in Figure A1 and Table A2 in the appendix, where each rule is semantically aligned with human-understandable driving norms such as lane keeping in work zones, avoiding barriers, and reasonable reactions to cones and signs. To partially mitigate the lack of annotations, we will add a qualitative analysis in the paper: we will sample representative scenarios and illustrate how the generated trajectories satisfy these rule constraints, thereby demonstrating, from a human observer’s perspective, both traffic-rule compliance and interpretability.
>
> Regarding passenger comfort (e.g., jerk and acceleration profiles), this is not the primary focus of the current work. In a typical autonomous driving stack, different modules are responsible for different aspects of behavior: our contribution focuses on high-level semantic reasoning and safe trajectory planning in complex work-zone scenarios, while fine-grained comfort optimization is usually handled by downstream trajectory smoothing and control modules. Nevertheless, we agree that incorporating explicit comfort metrics into our framework in future work would further improve the completeness of the overall system evaluation.
>
> > Q5: Fairness of baseline comparisons
>
> A5: We were initially somewhat unsure about the exact source of this concern, so we would like to clarify our experimental setup. Our primary baseline is a VLM (Qwen2.5-VL) fine-tuned on the ROADWork training set, which contains both successful and failed cases. REACT-Drive uses the same VLM backbone and is trained on the same ROADWork data, but instead of relying purely on end-to-end fine-tuning, we explicitly organize the failure information into a failure-case database with corresponding mitigation code. In other words, both methods have access to the same underlying information; the key difference is whether the failure information is kept implicitly in the model weights or explicitly modeled and exploited through retrieval and reasoning. From this perspective, we believe this is a fair comparison under a “same model, same data” setting that directly measures the gain brought by our framework.
>
> If the reviewer’s concern instead refers to our comparison with strong autonomous driving models such as DriveLM and RoboTron-Drive, we respectfully disagree that this is an unfair comparison in the sense implied. These models are typically trained on much larger and more diverse autonomous driving datasets, very likely including a wide variety of work-zone scenarios, and their papers explicitly emphasize strong generalization across many conditions. Therefore, we mainly use these systems as reference points to position our work within the broader landscape, rather than claiming a strictly one-to-one, equal-condition comparison with them.

---

> ### Author Response · Authors · 2025-11-20
> **Author Response to Reviewer bZho (3/5)**
>
> > Q6: The physical evaluation is too limited (open-loop, very few scenarios).
>
> A6: We agree that the current physical experiments are indeed limited in scale. Collecting real-world work-zone data is very costly in terms of both time and financial resources, which constrains the size of our evaluation. However, to better address this concern, we have expanded the physical experiments to cover 6 failure patterns, 23 scenarios, and 200 images, with the corresponding results shown below. We will include these additional results in the revised manuscript.
>
> | Model            | P1                     | P2                     | P3                     | P5                     | P7                     | P8                     | Total Avg              |
> |------------------|------------------------|------------------------|------------------------|------------------------|------------------------|------------------------|------------------------|
> |                  | ADE / FDE / CR         | ADE / FDE / CR         | ADE / FDE / CR         | ADE / FDE / CR         | ADE / FDE / CR         | ADE / FDE / CR         | ADE / FDE / CR         |
> | GPT4o            | 143.21 / 294.98 / 0.05 | 276.06 / 575.79 / 0.06 | 125.58 / 267.79 / 0.00 | 152.52 / 280.43 / 0.25 | 161.36 / 335.68 / 0.00 | 286.12 / 621.34 / 0.10 | 190.81 / 396.00 / 0.08 |
> | Finetune-VLM     | 104.75 / 206.26 / 0.02 | 336.47 / 657.53 / 0.06 | 109.62 / 217.66 / 0.00 | 129.27 / 252.82 / 0.21 | 109.44 / 217.10 / 0.00 | 317.83 / 697.67 / 0.10 | 184.56 / 374.84 / 0.07 |
> | DriveLM          | 165.40 / 341.53 / 0.03 | 368.69 / 728.15 / 0.06 | 167.84 / 351.33 / 0.00 | 226.78 / 436.79 / 0.04 | 161.36 / 335.68 / 0.00 | 374.86 / 738.60 / 0.00 | 244.16 / 488.68 / 0.02 |
> | SimLingo         | 184.89 / 327.17 / 0.02 | 358.07 / 705.67 / 0.00 | 179.89 / 314.67 / 0.00 | 203.05 / 360.87 / 0.08 | 143.33 / 248.59 / 0.00 | 363.76 / 690.28 / 0.00 | 238.83 / 441.21 / 0.02 |
> | RoboTron-Drive   | 152.30 / 328.40 / 0.05 | 355.20 / 710.50 / 0.00 | 155.60 / 338.20 / 0.00 | 173.40 / 320.30 / 0.08 | 128.90 / 302.10 / 0.03 | 360.50 / 720.40 / 0.00 | 220.98 / 453.32 / 0.03 |
> | REACT-Drive      | 52.88 / 104.84 / 0.00  | 61.46 / 145.32 / 0.00  | 64.74 / 112.38 / 0.00  | 64.04 / 87.40 / 0.15   | 61.33 / 92.77 / 0.00   | 68.76 / 101.39 / 0.15  | 62.20 / 107.35 / 0.05  |
>
> > Q7: Lack of comparison with classical planning algorithms
>
> A7: We believe there may be a misunderstanding here. In this paper, we intentionally do not provide a direct numerical comparison between VLM-based methods and classical or optimization-based planners (such as MPC), because such a one-to-one comparison is neither meaningful nor truly fair under the current setting. Traditional planning pipelines already exhibit well-known limitations in complex work-zone environments, which is precisely the motivation highlighted in our introduction through multiple real-world work-zone–related incidents.
>
> It is exactly due to these challenges that many industrial systems and car manufacturers have recently begun exploring VLM-based approaches. Our work operates within this VLM paradigm, focusing on improving the performance of VLM planners specifically in work-zone scenarios, rather than re-evaluating or replacing classical planners themselves. We will clarify this motivation and scope more explicitly in the revised manuscript to prevent similar misunderstandings.
>
> > Q8: System complexity and end-to-end latency
>
> A8: We believe there is a misunderstanding regarding the “heavy-weight design” and the latency of our system. REACT-Drive is explicitly divided into an offline stage and an online stage. The most time-consuming components, including abnormal subgraph analysis, VLM-based code generation, and code verification, are all performed in the offline stage to build the failure-case mitigation database.
>
> During the online stage, the vehicle only needs to perform lightweight scene-graph construction, retrieval, and execution of code snippets that have already been fully validated offline, which does not introduce significant computational overhead. As shown in Fig. 3, the online inference time of REACT-Drive is 0.58s in our implementation, which is comparable to other VLM-based approaches.
>
> Furthermore, as discussed in Q1, the perception outputs required for constructing the scene graph, such as work-zone object detections and depth information, are typically already available in modern autonomous driving stacks. Therefore, our framework does not introduce additional heavy perception modules beyond what existing systems already use. Also, in real deployments, the inference time can be even faster and more accurate (with more advanced modality like lidar).

---

> > ### Author Response · Authors · 2025-11-20
> > **Author Response to Reviewer bZho (4/5)**
> >
> > > Q9: Subjectivity and reliability of failure-pattern summarization
> >
> > A9: We agree that the construction of the eight failure patterns inevitably involves a certain degree of subjectivity. In practice, we first perform clustering (with the number of clusters selected using the elbow method), after which multiple domain experts jointly review, merge, and name the clusters. Although this process is not fully automated, our goal is to obtain a set of broadly representative failure behaviors rather than a highly fine-grained taxonomy that only applies to a small number of cases.
> >
> > In addition, we analyze the failure cases that are not covered by the current pattern set. This includes examining how many cases are omitted and evaluating how the underlying VLM performs on these uncovered scenarios. This helps us understand both the limitations of the current pattern set and the robustness of the system when patterns are incomplete.
> > Finally, we emphasize that introducing failure patterns and the mitigation mechanism does not reduce planning performance in normal (non-failure) scenarios. When no relevant failure pattern is retrieved, the system naturally falls back to the original VLM behavior. Therefore, even if the current set of patterns is not perfectly exhaustive or mutually exclusive, it still provides clear benefits in a wide class of challenging cases while preserving baseline performance in ordinary situations.
> >
> >
> >
> > > Q10: Online updating of the failure-case database and multi-agent interactions
> >
> > A10：We would like to clarify that the failure-case database is not updated during online operation. It is fully constructed and validated in the offline stage, which ensures that all mitigation codes are safety-checked before deployment. During real-world driving, the vehicle only retrieves from a fixed, pre-verified database. Although the REACT-Drive framework already exhibits a certain level of transferability, we acknowledge that, in practice, different regions can maintain separate failure-case databases, which can further improve accuracy and regional relevance.
> >
> > Regarding multi-agent interactions, we emphasize that this capability is typically realized by multiple components in a full autonomous driving stack, including perception, prediction, behavior planning, and control, rather than by the trajectory planning module alone. Our work focuses on enhancing trajectory reasoning in work-zone scenarios within a VLM-based framework. The ROADWork dataset itself lacks rich multi-agent information, so performing fully interaction-aware reasoning would require more comprehensive upstream modules and datasets, which go beyond the scope of this paper. Nevertheless, the architecture of REACT-Drive is modular, and when more comprehensive datasets and stronger upstream interaction models are available, our system can be naturally extended to support multi-agent reasoning.
> >
> >
> > > Q11: Dataset limitations and generalization beyond ROADWork
> >
> > A11：To the best of our knowledge, ROADWork is the first and only open-source dataset specifically designed for traversing construction zones with paired trajectory annotations. Although it lacks certain elements such as full 3D information and rich multi-agent interactions, it remains the most comprehensive publicly available benchmark for work-zone scenarios, which is why we chose it as our primary evaluation dataset.
> >
> > In addition, as described in the “Transferability” section of the paper, we conducted physical experiments in real-world work zones located in regions entirely different from those where ROADWork was collected. These experiments serve as a preliminary test of generalization beyond the dataset. The results show that REACT-Drive maintains strong performance even in unseen areas, suggesting that it possesses a reasonable level of real-world generalizability in work-zone scenarios.

---

> > > ### Author Response · Authors · 2025-11-20
> > > **Author Response to Reviewer bZho (5/5)**
> > >
> > > > Q12: Lack of coverage of extreme weather and nighttime conditions
> > >
> > > A12: We acknowledge that the current experiments do not systematically include extreme weather or nighttime scenarios. This limitation primarily stems from the ROADWork dataset itself. As noted in Q11, ROADWork is currently the most comprehensive open-source work-zone dataset with trajectory annotations, but it does not contain diverse adverse-weather or low-light conditions.
> > >
> > > It is important to note that many recent studies have shown that VLMs exhibit reasonable robustness under challenging visual conditions, such as illumination changes and moderate adverse weather [1,2]. When stronger upstream perception modules are used, for example, multi-sensor fusion or enhanced night-vision/low-light models, REACT-Drive can naturally benefit from these improvements as well. To demonstrate it, we collect two nighttime scenes similar to our P1 and P8 from the recently released open-waymo dataset. Results show that in image (972,1079), the (AvgDET, AvgFDE, CR) metrics for the two scenes were (24.08, 39.44, 0) and (40.73, 82.38, 0). This demonstrates that our work also performs well in nighttime conditions. We will include these results into the revised version.
> > >
> > > We also emphasize that extremely low-visibility conditions, such as dense fog, blizzards, or severe nighttime glare, typically fall outside the Operational Design Domain of most autonomous driving systems and pose significant risks even for human drivers. In real deployments, such scenarios should be handled through system-level safety strategies (e.g., operation restrictions, active degradation/exit), rather than relying solely on the trajectory planning module.
> > > We will make this dataset limitation and the system’s applicable operating conditions clearer in the revised manuscript.
> > >
> > >
> > > > Conclusion:
> > >
> > > Overall, many of the concerns raised by the reviewers indeed stem from the inherent limitations of the ROADWork dataset, and these limitations are difficult to fully resolve within the time and resource constraints of this submission. Nevertheless, we have conducted additional experiments wherever feasible and reported the corresponding results above. If the reviewers have further questions, we would greatly appreciate additional comments and will do our best to supplement experiments within the rebuttal period.
> > >
> > > At the same time, we recognize that ICLR encourages openness and collaboration. If any industry teams or research groups are interested in our paper and this response, we would be very happy to collaborate after the review process to expand work-zone datasets and jointly advance VLM-based planning research. We believe this direction is of substantial importance for improving safety in complex long-tail driving scenarios.
> > >
> > > Authors
> > >
> > > Reference:
> > >
> > > 1. Xu et al., “Towards Real-World Adverse Weather Image Restoration: Enhancing Clearness and Semantics with Vision-Language Models”, ECCV 2024.
> > >
> > > 2. Yang et al., “Language-driven All-in-one Adverse Weather Removal”, CVPR 2024.

---

> ### Author Response · Authors · 2025-11-23
> **Summary of our revision based on Reviewer bZho's comments**
>
> We sincerely appreciate your insightful comments and constructive feedback, we have uploaded a revised manuscript to solve your valuable comments.
>
> bZho-Q1
> > Perception errors from YOLOv12 / MiDaS and safety risks
>
> We have added a discussion in Appendix Section C, which conveys the same points as provided in our previous response, with some refinement for clarity. Please refer to page 14, lines 734–744 of the revised manuscript.
>
> bZho-Q3
> > Limitations of pixel-space metrics
>
> We have also added a discussion in Appendix Section C, where we incorporated and expanded the points from our response. Please refer to page 14, lines 745–753 of the revised manuscript.
>
> bZho-Q6
> > The physical evaluation is too limited (open-loop, very few scenarios).
>
> In response to your comments, we promptly expanded the physical experiments by collecting additional data and conducting more comprehensive evaluations. The updated results can be found on page 9, lines 503–517. Moreover, Appendix Table A2 presents the results for each physical experiment pattern individually.
>
>
> bZho-Q12
> >Lack of coverage of extreme weather and nighttime conditions
> Consistent with our response, we have added a corresponding discussion on the performance under nighttime conditions in Appendix Section C; please refer to page 14–15, lines 754–770 of the revised manuscript. In addition, a visualization example is provided in Figure A1(a).
>
> bZho-Q2, Q4, Q5, Q7, Q8, Q9, Q10, Q11
>
> For these questions, the issues appear to stem from some misunderstandings, which we believe have been fully clarified in our response. We feel that these points may not be suitable to include in the manuscript, as doing so might distract from the main contributions.
> If you have further concerns, we would be glad to continue the discussion. Otherwise, we hope that the clarifications provided can help resolve the remaining doubts. If the responses adequately address your concerns, we would greatly appreciate your consideration in reflecting this in your evaluation, as it is very important for the progression of our work.

---

> ### Author Response · Authors · 2025-11-26
> **Official Comment by Authors**
>
> Dear reviewer,
>
> We would be very happy to discuss any further questions before the end of the discussion phase. In particular, please let us know if your questions around the mitigate code generation in our method, the extremely weather evaluation and physical evaluation have been resolved by our edits.
>
> We truly understand that you may also be busy with your own work during this period, but we would greatly appreciate it if you could spare a moment to share any remaining thoughts.
>
> Thanks again for your comments which helped a lot for revising our manuscript.
>
> Authors.

---

> ### Author Response · Authors · 2025-11-27
> **Official Comment by Authors**
>
> Dear reviewer bZho,
>
> Sorry to bother you again. Since we are now approaching the last week of the discussion period, we would be truly grateful if you could take another look at our responses when you have a moment. Your feedback would be extremely helpful for us.
>
> Thanks again for your comments which helped a lot for revising our manuscript.
>
> Authors.

---

### Meta-Review · Area_Chair_ueKB · 2026-01-07

**Summary:**

Reviewers praised:
- The method's systematic evaluation of VLM planning performance on the ROADWork dataset and the resulting insights.
- The paper's clarity.
- The experimental thoroughness.

Reviewers were concerned about:
- The strength of the auto-labeling models used in the REACT-Drive framework
- The decision to generate and execute Python code at runtime.
- Pixel-wise ADE and FDE values compared to metric (e.g., in meters) values.
- Missing evaluations of trajectory quality (e.g., jerk and acceleration values, lane discipline).
- Different training methodologies between baselines and the proposed approach.
- Evaluation is limited to one dataset, the open-loop setting, and small scale.
- Weak scenario coverage.

**Reviewer Concerns:**

- The strength of the auto-labeling models used in the REACT-Drive framework: They can be swapped out for better ones if desired
- The decision to generate and execute Python code at runtime: Generation is only done offline.
- Missing evaluations of trajectory quality (e.g., jerk and acceleration values, lane discipline): Not possible with the ROADWork dataset, attempted with the real-world-collected data experiments.
- Weak scenario coverage: Expansion with nighttime scenes from the Waymo dataset.

Importantly, "evaluation is limited to one dataset, the open-loop setting, and small scale". This is the dominant outstanding concern, as I see it.

**Reviewer Scores:**

- Reviewer yKor indicated that their core concerns were addressed, so I believe they would have changed their score to a 6.

---

### Decision · Program_Chairs · 2026-01-26

Reject